

**Evaluation of the NAQFC Driven by the NOAA Global Forecast**
**System Version 16: Comparison with the WRF-CMAQ Downscaling**
**Method During the Summer 2019 FIREX-AQ Campaign**
Youhua Tang[1,2], Patrick C. Campbell[1,2], Pius Lee[1], Rick Saylor[1], Fanglin Yang[3], Barry Baker[1],
Daniel Tong[1,2], Ariel Stein[1], Jianping Huang[3,4], Ho-Chun Huang[3,4], Li Pan[3,4], Jeff McQueen[3],
Ivanka Stajner[3], Jose Tirado-Delgado[5,6], Youngsun Jung[5], Melissa Yang[7], Ilann Bourgeois[8,9],
Jeff Peischl[8,9], Tom Ryerson[9], Donald Blake[10], Joshua Schwarz[9],
Jose-Luis Jimenez[8], James Crawford[11], Glenn Diskin[7], Richard Moore[7], Johnathan Hair[7], Greg
Huey[11], Andrew Rollins[9], Jack Dibb[12], Xiaoyang Zhang[13]
1. NOAA Air Resources Laboratory, College Park, MD, USA.
2. Center for Spatial Information Science and Systems, George Mason University, Fairfax, VA,
USA.
3. NOAA National Centers for Environmental Prediction, College Park, MD, USA
4. I.M. Systems Group Inc., Rockville, MD, USA
5. Office of Science and Technology Integration, NOAA National Weather Service, Silver
Spring, MD, USA
6. Eastern Research Group Inc, USA
7. NASA Langley Research Center, Hampton, VA, USA
8. Cooperative Institute for Research in Environmental Sciences, University of Colorado
Boulder, Boulder, CO, USA
9. NOAA Chemical Sciences Laboratory, Boulder, CO, USA
10. Department of Chemistry, University of California at Irvine, Irvine, CA, USA
11. School of Earth and Atmospheric Sciences, Georgia Institute of Technology, Atlanta, GA,
USA
12. Earth Systems Research Center, University of New Hampshire, Durham, NH, USA
13. Department of Geography & Geospatial Sciences, South Dakota State
University, Brookings, SD, USA
**Correspondence**: Youhua Tang (youhua.tang@noaa.gov)
**Abstract**
The latest operational National Air Quality Forecasting Capability (NAQFC) has been advanced
to use the Community Multi-scale Air Quality (CMAQ) model version 5.3.1 with CB6 (carbon
bond version 6)-Aero7 (version 7 of the aerosol module) chemical mechanism and is driven by
the Finite Volume Cubed-Sphere (FV3)-Global Forecast System, version 16 (GFSv16). This has
been accomplished by development of the meteorological preprocessor, NOAA-EPA
Atmosphere-Chemistry Coupler (NACC), which is adapted from the existing Meteorology-
Chemistry Interface Processor (MCIP). Differing from the typically used Weather Research and
Forecasting (WRF)/CMAQ system in the air quality research community, the interpolation-based
NACC can use various meteorological output to drive CMAQ (e.g., FV3-GFSv16) even though



they are in different grids. Here we compare and evaluate GFSv16-CMAQ vs. WRFv4.0.3-
CMAQ using observations over the contiguous United States (CONUS) in summer 2019. During
this period, the Fire Influence on Regional to Global Environments and Air Quality (FIREX-AQ)
field campaign was performed and we compare the two models with airborne measurements
mainly from the NASA DC-8 aircraft. The GFS-CMAQ and WRF-CMAQ systems have overall
similar performance with some differences for certain events, species and regions.  The GFSv16
meteorology tends to have stronger planetary boundary layer height diurnal variability (higher
during daytime, and lower at night) than WRF over the U.S. Pacific coast, and it also predicted
lower nighttime 10-m winds. In summer 2019, GFS-CMAQ system showed better surface $O_3$
than WRF-CMAQ at night over the CONUS domain; however, their PM2.5 predictions showed
mixed verification results: GFS-CMAQ yielded better mean bias but poorer correlations over the
Pacific coast. These results indicate that using global GFSv16 meteorology with NACC to
directly drive CMAQ via the interpolation is feasible and yields reasonable results compared to
the commonly-used WRF downscaling approach.

## 1. Introduction

Traditionally, mesoscale meteorological models such as the Weather Research and Forecasting
Model (WRF) (Powers et al., 2017) are used as the meteorological drivers for air quality models
(AQMs) on the same ("native") model grid, such as Community Multiscale Air Quality Model
(CMAQ) (Byun & Schere, 2006). The NOAA National Weather Service's (NWS) National Air
Quality Forecasting Capability (NAQFC) has historically used a different approach, in which the
hourly meteorological outputs from prior operational models, such as North American Mesoscale
Model (NAM), need to be interpolated to the AQM grid to drive its air quality prediction. Prior
to this work, a "PREMAQ" coupler (Otte et al, 2004) combined both meteorological processing
and Sparse Matrix Operator Kernel Emissions (SMOKE) (Houyoux et al., 2000) processes, such
as point source plume rise effects. However, since the release of CMAQ version 5, the
meteorology-dependent plume rise, sea salt, and dust emission processes are included as inline
modules in CMAQ, and thus the corresponding emission processes are no longer needed in
PREMAQ. Furthermore, PREMAQ has no built-in interpolator, and thus relied on external
interpolators to remap the non-native-grid meteorological inputs, such as NAM, to the targeted
CMAQ domain, though it did perform vertical layer collapsing/interpolation to reduce layers.
The interpolation approach allows for more flexibility in using different meteorological drivers
(i.e., besides just WRF) for CMAQ; however, there is potential to raise mass-consistency issues
between models. It should be noted that the mass-consistency issues may also exist using native-
grid couplers (Byun, 1999a, 1999b), which can stem from the mass-inconsistent meteorological
inputs or due to the temporal interpolation of the input data. The well-developed offline AQMs,
such as CMAQ, have already considered such mass-consistency treatments using different
meteorological inputs (Byun et al., 1999c).
To upgrade the NAQFC system with the latest CMAQ chemistry and NOAA operational
meteorology, we developed an updated interpolation-based meteorological coupler, the NOAA-
EPA Atmosphere-Chemistry Coupler (NACC) (Campbell et al., 2022) adapted from the U.S.



EPA's Meteorology-Chemistry Interface Processor (MCIP) version 5 (Otte and Pleim, 2010;
https://github.com/USEPA/CMAQ). The NACC system replaced PREMAQ, and effectively
couples the Finite-Volume Cubed-Sphere (FV3) Dynamical Core - Global Forecast System
version 16 (GFSv16) (Yang et al., 2020; Harris et al, 2021) to CMAQ v5.3.1 (hereafter referred
to as GFS-CMAQ). Campbell et al. (2022) described the development and application of the
GFS-CMAQ system using NACC (in their work referred to as "NACC-CMAQ") and a
comprehensive comparison between the current (GFS-CMAQ since July 20, 2021) and previous
(NAM-CMAQv5.0.2) operational NAQFC model performances.
In this study, we analyze the impacts of the meteorological model drivers, and compare GFS-
CMAQ using NACC interpolation to the commonly-used downscaled, native-grid WRF-CMAQ
application and its impact on air quality predictions. Yu et al. (2012a, 2012b) had previously
compared the CMAQ performance driven by WRF-NMM and WRF-ARW during the 2006
TexAQS/GoMACCS field campaign, and found that the NMM-CMAQ and ARW-CMAQ
showed overall similar performance with some differences for certain events, chemical species,
and regions. Similarly, this study focuses on the comparison of GFS-CMAQ versus WRF-
CMAQ (see Section 2, Methodology), and verifies the model performance against the aircraft
observations from the Fire Influence on Regional to Global Environments and Air Quality
(FIREX-AQ) field experiment during summer 2019 (Section 3). Surface verification is also
performed using AIRNow data for August 2019 (Section 4), serving as a benchmark for the new
NAQFC versus the traditional WRF-CMAQ used in the air quality modeling community.
## 2. Methodology
Here we compare the two CMAQ (version 5.3.1) runs driven by the interpolated GFSv16
meteorology (GFS-CMAQ) and WRF downscaled meteorology (WRF-CMAQ). All other
settings, such as emission and lateral boundary conditions are the same.  The meteorology-
related physics is discussed in the following sections to address the models' performance
discrepancies. Both the GFS-CMAQ and WRF-CMAQ simulations are run from a period
covering 12 July – 31 August, 2019, each using the last 10 days in July as the model spin-ups
that are not included in the analyses.
**2.1 GFS Meteorological Inputs**
The GFSv16 is the current operational global forecast system in NOAA/NCEP using FV3
dynamical core.  Its detailed configuration can be found in Campbell et al. (2022) and Yang et al.
(2020). Compared to the previous version (v15), GFSv16 updated many physical schemes (Table
1) and added the parameterization for subgrid scale nonstationary gravity-wave drag. To use the
GFS's meteorology to drive CMAQ, a meteorological coupler, NACC, is developed (Campbell
et al., 2022). Differing from the original MCIP which was developed to process WRF/ARW
meteorology for CMAQ, the NACC coupler interpolates non-native-grid meteorology to a user-



defined grid and has parallel processing capability, which drastically reduces its run time for
operational forecasts (Campbell et al., 2022). Currently, NACC's horizontal interpolation
employs two methods: bilinear and nearest-neighbor. In this study, we use the nearest-neighbor
method to categorical (discontinuous) variables that include land use types, vegetation fraction,
terrain elevation, Monin–Obukhov length, friction velocity, and soil temperatures, while the
bilinear interpolation is used for mainly smoothly varying (continuous) meteorological variables
that include wind fields, temperature, pressure, and specific humidity. The CMAQ model is
defined in the Arakawa C-grid (Arakawa and Lamb, 1977), and thus the GFSv16 horizontal wind
components (U, V) need to be interpolated to the perpendicular cell faces instead of the cell
center (Otte and Pleim, 2010) after rotation to the defined map projection. The scalar variables
are defined in the target grid cell center, and thus their GFSv16 interpolations are more
straightforward. The NACC coupler can either use the native layers or collapse (i.e., interpolate)
to a set number of user-defined vertical layers for CMAQ use. The GFSv16 has 127 vertical
layers with global coverage in 13 km horizontal resolution, where the targeted domain is a 12×12
km Contiguous United States (CONUS) with 35 vertical layers (Campbell et al, 2022). Here we
use 24-hour GFSv16 forecasts starting at 12 UTC each day.
Most variables needed by CMAQ are directly interpolated from the GFSv16 outputs. The NACC
processor has options to calculate diagnostic variables, such as planetary boundary layer (PBL)
height, if they are needed. In this study, we use the interpolated GFSv16's PBL height instead of
the diagnostic one. It also has an option to import the externally provided land-surface variables.
Here we import updated 2018–2020 climatological averaged leaf area index (LAI) and NOAA
near-real-time (NRT) greenness vegetation fraction (GVF) from satellite-based Visible Infrared
Imaging Radiometer Suite (VIIRS) retrievals (Campbell et al., 2022). The updated satellite-based
LAI and GVF impact CMAQ's biogenic emissions and dry deposition processes, which were
described in detail in Campbell et al. (2022).
**2.2 WRF Meteorology**
To compare with GFSv16 meteorology processed by NACC, a corresponding WRF version 4.0.3
(Skamarock et al, 2021) simulation is run covering the NAQFC's native grid, which is a 12 km
horizontal resolution, Lambert conformal map projection over CONUS.  Table 1 shows the WRF
configuration, which is commonly employed in CONUS meteorological and air quality studies in
the community, versus the current NOAA/NWS operational version of GFSv16. In contrast to
GFSv16, which is a global model that uses the NOAA/NCEP's Global Data Assimilation System
(GDAS) (https://www.emc.ncep.noaa.gov/data_assimilation/data.html) for its initial conditions
and runs on its own global dynamics and physics without any other constraints, the regional
WRF simulation uses downscaled GFSv16 for its initial conditions. Furthermore, WRF also uses
downscaled lateral boundary conditions taken from GFSv16 every 6 hours. Here WRF runs
continuously after spin-up and we have enabled the four-dimensional data assimilation (FDDA)





for the u- and v-component winds, temperature, and humidity (Table 1) every 6 hours, thus
nudging towards GFSv16.
WRF and GFSv16 have similar settings for the land surface model, surface layer and radiation
schemes; however, their microphysics and PBL schemes are different (Table 1). Compared to the
35-layer WRF with a 100 hPa domain top, GFSv16 has a much higher domain top (0.2 hPa) and
127 vertical layers, which are collapsed by NACC to 35 sigma layers up to 14 km for CMAQ.
We use NACC (inherited from MCIP version 5.0) to process WRF hourly meteorology, while
maintaining the vertical layer structure. Thus, in contrast to GFS-CMAQ, the WRF-CMAQ
system uses the native grid without interpolation.
**2.3 CMAQ Configuration**
Here CMAQ version 5.3.1 (Appel et al., 2021) is used with the Carbon Bond 6 version r3
(CB6r3; Yarwood et al., 2010, 2014; Luecken, et al., 2019) chemical mechanism and Aero7
treatment of secondary organic aerosols (CB6r3_AE7_AQ).  CMAQ 5.3.1 includes a series of
scientific updates from the previous version (Appel et al., 2021), including the updated air-
surface exchange and deposition modules, which showed significant impact on ozone prediction
compared to the previous NAQFC (Campbell et al., 2022). We also include the bi-directional
$NH_3$ (BIDI-$NH_3$) exchange model for $NH_3$ surface fluxes. An updated Biogenic Emissions
Landuse Dataset v5 (BELD5) is used in this study to drive the inline Biogenic Emissions
Inventory System (BEIS) version 3.61. The anthropogenic emissions are provided by the
National Emissions Inventory Collaborative (NEIC) with base year 2016 version 1 (NEIC 2019).
We replace the U.S. EPA default CMAQ dust emissions model with a novel inline windblown
dust model known as "FENGSHA" (Fu et al., 2014; Huang et al., 2015; Dong et al., 2016).
Campbell et al. (2022) include the details of the CMAQ 5.3.1 configuration for this study.
We have updated the wildfire emissions system in CMAQv5.3.1 based on the Blended Global
Biomass Burning Emissions Product (GBBEPx) (Zhang and Kondragunta, 2006; Zhang et al.,
2011).  The GBBEPx uses satellite-detected fire radiative power (FRP) to estimate wildfire
smoke emissions for a number of species: CO (carbon monoxide), NOx (nitrogen oxides), $SO_2$
(sulfur dioxide), elemental carbon, total organic aerosols, and PM2.5. The satellite FRP is
estimated from satellite brightness temperature anomaly, and the GBBEPx processor assumes
that the wildfire emissions are proportional to the FRP over certain land use type in certain
regions. The GBBEPx emissions are based on polar orbiting satellites: MODIS (Aqua and Terra
satellites) and VIIRS (Suomi-NPP and NOAA-20 satellites) instruments, which are updated once
per day. A wildfire emission preprocessor converts the GBBEPx emissions to CMAQ-ready
input files using emission speciation and diurnal profiles (high during daytime and low at night)
(adopted from U.S. EPA-based profiles) (Baker et al., 2016), and a daily scaling factor. Here we
classify the wildfire into either a long-lasting fire (longer than 24 hours) or short-term fire
(shorter than 24 hours) based on land use types and regions.  Only the fires west of 110°W that





have a model grid cell total forest fraction > 0.4 are assumed to be long-lasting fires, which incur
daily scaling factors of 1, 0.25, 0.25 for days 1, 2 and 3, respectively. All other short-term
GBBEPx fires are assumed to have smoke emissions for 24 hours (i.e., day 1 only).  CMAQ
treats wildfire emissions as point sources that undergo inline plume rise to distribute the smoke
vertically. The default CMAQ plume rise used here is based on Briggs (1965), which is driven
by fire heat flux (converted from FRP with a ratio of 1) and fixed burning area (assumed to be
10% of the 0.1°×0.1° grid cell).
## 3.  Model Evaluations over the U.S. for August 2019
To first gain a general picture and compare the overall GFS-CMAQ and WRF-CMAQ model
performances, in this section we evaluate near-surface meteorological and air quality predictions
during the FIREX-AQ August 2019 period against NOAA's METeorological Aerodrome Report
(METAR; https://madis.ncep.noaa.gov/madis_metar.shtml) and the U.S. EPA's AirNow
(https://www.airnow.gov/) observation networks.
**3.1 Domain-Wide Meteorology against the METAR Network**
Figure 1 shows the mean bias (MB) of GFS and WRF predicted surface meteorological variables
compared to METAR data during August, 2019. Both meteorological models have a cool bias
over the Western and Northeastern United States, and a warm bias over the western Rocky
Mountain region and Southeastern United States (Figure 1a, 1b). Similar temperature predictions
are expected since WRF uses the FDDA method nudging toward GFS data. However, GFS tends
to be cooler than WRF over the Rocky Mountains and in the central and northeastern USA due
to their different dynamics and physics.   The GFSv16 cold bias in the lower troposphere is
impacted by excessive evaporative cooling from rainfall (personal communication with
NOAA/NCEP). Campbell et al. (2022) had detailed discussions about GFSv16 biases.
Both GFSv16 and WRF models have similar and rather significant dry biases for specific
humidity (SH) predictions across CONUS (Figure 1c, 1d). Qian et al (2020) investigated this
common dry bias in many models, and found that neglecting an irrigation contribution could
cause this dry bias. GFS has widespread dry biases (Campbell et al. 2022) and WRF has similar
dry biases, too as it is nudged toward GFS. There are some noticeable differences for certain
regions. For instance, WRF has less dry bias over Southern Texas than GFS.
Both models underestimate the mean 10-m wind speeds compared to METAR stations over the
western U.S. WRF has stronger underpredictions over the Rocky Mountains and overpredictions
over northeastern U.S., while GFS has stronger underpredictions over the Appalachian
Mountains and overpredictions over Texas and Oklahoma. GFSv16's operational verification
also (https://www.emc.ncep.noaa.gov/gmb/emc.glopara/vsdb/v16rt2/g2o/g2o_00Z/index.html)
shows that it tends to underpredict the 10-m wind over the western U.S. during both daytime and





nighttime, but shows overprediction over the eastern U.S. Besides the difference of physical
schemes, etc. (Table 1), other possible reasons causing this surface wind difference could be
effect of the gravity-wave drag (GFSv16 includes it, but the WRF run here does not), and
vertical resolution (GFS's 127 layers versus WRF's 35 layers). Some studies (Skamarock et al,
2019) revealed the necessity of fine vertical resolution for atmospheric simulations, especially
within the PBL, near tropospheric top, and during convective events. Insufficient vertical
resolution could also cause plume dilution on chemical transport modeling (Zhuang et al., 2018).
The gravity-wave drag is also known to produce synoptic scale body forces on the atmospheric
flow over irregularities at the earth's surface such as mountains and valleys, and uneven
distribution of diabatic heat sources associated with convective systems (Kim et al., 2003). Its
parameterization is needed for large-scale models.
There is strong regional variability in the monthly mean PBL height differences between GFS
and WRF during daytime (represented by 18 UTC) and nighttime (represented by 06 UTC)
(Figure 2). During daytime, GFS has a higher PBL height compared to WRF over the U.S.
Pacific coast, northern Rocky Mountains, northeastern and southeastern U.S., but it becomes
lower over the central U.S. (e.g., Texas, Oklahoma, and Kansas). At night, however, most of
these regional differences between GFS and WRF are reversed.  This diurnal difference is
mainly driven by the different PBL schemes employed in GFS (Han and Bretherton, 2019) and
WRF (i.e., YSU) and the associated other physical suites, including the land surface data. The
GFS's PBL height has a strong diurnal variation over these regions, including the western and
northeastern U.S. in the summer, including a sharp rise and collapse after sunrise and sunset,
respectively (Campbell et al., 2022). The strong PBL diurnal variation has significant effects on
the air quality predictions in GFS-CMAQ.
**3.2 Evaluation of Regional Meteorology and Air Quality against the AirNow Network**
The U.S EPA AirNow network provides hourly observations of near-surface ozone, fine
particulate matter (PM2.5), and meteorology. Campbell et al. (2022) showed detailed verification
of GFS-CMAQ with the surface AIRNow data. Here we focus on the difference between the
interpolation-based GFSv16 versus WRF downscaling and the impacts on meteorological and
chemical model performances. Figure 3 shows a comparison of these two models over two
specific regions, the U.S. West (CA, OR and WA) and Northeast states (CT, DE, MA, MD, ME,
NH, NJ, NY, PA, RI, VT and District of Columbia) (Figure S1), where the two models had
relatively large differences for some meteorological variables. GFS and WRF had very similar 2-
m temperatures over the Pacific coast states: Washington, Oregon and California, and both of
them had similar cool bias (around 1K), R and RMSE (Figure 3a).  However, these two models
had significant differences for10-m wind speed prediction over the Pacific coast (Figure 3c),
where WRF overpredicted the wind speed, especially at night and in later August. Most AIRNow
stations are located near urban or suburban areas, which generally have weaker 10-m wind speed
than those at the METAR aviation weather stations near airports. For this reason, although





Figure 1e and 1f shows that GFS and WRF underpredict monthly-mean wind speed over the
METAR stations in the West, they still tend to overpredict AIRNow wind (Figure 3c), especially
for the WRF 10-m wind speed at night. Considering that the model grid cells represent 12×12
$km^2$ averages, the true model-observation comparisons likely fall somewhere between the
urban/suburban AIRNow stations and METAR stations, depending on the land use fractions of
each grid. Obviously the observation representation characteristics could affect the verification
results. Compared to AIRNow stations, GFSv16 has overall better scores for surface wind speed
predictions over the U.S. West, where the WRF's larger surface wind speed overprediction is
associated with its PBL height predictions (Figure 3e, 3f). During the nighttime, GFS has a lower
PBL height (10–50% lower than WRF) and weaker vertical mixing, which brings less
momentum from the upper layers to the surface, which led to lower nighttime wind and better
agreements with the AIRNow wind-speed observation.
Over the northeast, the mean bias (MB) of GFS temperature was about -1K, while the WRF has
a smaller, slightly positive MB of about 0.22K (Figure 3b). However, the GFS's temperature
prediction has a better correlation coefficient, R, and RMSE, implying that it better captures
some events, such as the 28–29 of August. Both models overpredict 10-m wind speeds in the
northeast, but the GFS model yields better results due to a slightly lower PBLH at night (Figure
3f) than WRF that had significant ovepredictions, especially during 25–29, August (Figure 3d)
when the tropical storm Erin approached this region. Especially on 28 of August, when the storm
was centered near the east coast of North Carolina, the WRF run significantly underpredicts 2-m
temperature (Figure 3b) and overpredict 10-m wind speed (Figure 3d), implying that the some
WRF settings lead to relatively large surface prediction bias for the storm weather, such as its
relatively coarse vertical resolution compared to the 127-layer GFS model.
Figures 4a and 4b show the ozone predictions of the two models over these two regions, and
GFS-CMAQ yields predominantly lower $O_3$ than WRF-CMAQ, especially at night. Over the
northwest, the lower ozone in GFS-CMAQ is associated with their PBL height difference. First,
with a certain dry deposition velocity between the models, it is easier to deplete ozone given the
smaller volume of a shallower PBL. Second, the thinner PBL results in higher NOx
concentrations and ozone titration rates near NOx source regions, and consequently lower ozone
there at night. Last, the lower PBL leads to weaker vertical mixing and downward transport of
ozone from the residual-layer at night (Caputi et al, 2019). All these factors contributed to the
lower nighttime ozone of GFS-CMAQ compared to WRF-CMAQ. Since GFS-CMAQ already
underpredicts ozone due to combined meteorological factors, such as the temperature
underprediction (Figure 4a), the GFS-CMAQ's further ozone reduction (possibly due to its lower
PBLH at night) exacerbates its low bias. However, over the Northeast, the similar impacts help
the GFS-CMAQ yield much better MB due to its better agreement with the observed nighttime
low ozone over the Northeast. Over the entire CONUS domain, the situation is similar: for an
average August 2019, the GFS-CMAQ has a lower ozone MB (1.1 ppb) compared to WRF-



CMAQ (4.7 ppb). Figure 5 shows that both models have similar daytime ozone prediction over
CONUS. However, GFS-CMAQ better captures low nighttime ozone over the U.S. East than
WRF-CMAQ (Figure 5c, 5d).
GFS-CMAQ has substantially higher PM2.5 mean concentrations over the U.S. West, but lower
over the U.S. Northeast compared to WRF-CMAQ (Figures 4c, 4d). These model differences are
also related to their interpolated GFSv16 versus downscaled WRF meteorological drivers.
Because both models use the same emissions under relatively clean background conditions in the
west (i.e., prevailing westerly flow from the Pacific Ocean), the PBL and wind speed differences
have significant impacts on their near-surface pollutant concentrations, especially at night. Both
models show strong PM2.5 diurnal variation (high at night and low during daytime), driven by
the meteorological diurnal variation (e.g. PBL), which overcomes the emission diurnal variation
(usually high during daytime and low at night). Compared to WRF-CMAQ, GFS-CMAQ has
lower nighttime PBL height and weaker wind speed at night, which leads to weaker vertical
mixing and venting, and increases the pollutant concentrations near the surface and yields higher
surface PM2.5 over the U.S. West (Figure 4c). Its higher surface PM2.5 could also result in
stronger local dry deposition. In contrast to local vertical mixing and venting effects on PM2.5
discussed above, there are strong (and potentially counterbalancing) impacts of model PBL and
horizontal wind speed differences on downstream PM2.5 concentrations at night. WRF-CMAQ's
deeper PBL and stronger wind speeds at night (Figures 3c–3f) tends to transport aerosols and
their precursors more efficiently downstream via the dominant advection pathway. Figure 6
shows that these monthly mean background PM2.5 differences appear in East of Rocky
Mountain (WRF-CMAQ is about 2 $\mu g/m^3$ higher) during both daytime and nighttime. This effect
is very prominent in the Northeast region. Although both models predicted similar PM2.5
magnitude over the U.S. Northeast, GFS-CMAQ yields the overall PM2.5 underprediction, and
its monthly-mean PM2.5 is 2.6 $\mu g/m^3$ lower than the WRF-CMAQ prediction (Figure 4d).
Especially during 01–09 August, WRF-CMAQ had about 4 $\mu g/m^3$ higher surface PM2.5
background than that of GFS-CMAQ. In this case, the WRF-CMAQ model has a better
agreement with observations (Figure 4d). It is possible that the GFS-CMAQ's nighttime PBL
heights (wind speeds) are too shallow (weak) in this case, which does not allow enough transport
of pollutants to the downstream (Eastern USA). Overall, GFS-CMAQ and WRF-CMAQ have
mixed performances for PM2.5 predictions during the August 2019 period: GFS-CMAQ has
better PM2.5 prediction over the U.S West, and WRF-CMAQ yields better results over east of
Rocky mountain (Figure 6).
**4   Model Comparisons against the FIREX-AQ Aircraft Data**
From late July to early September, 2019, the joint NOAA-NASA FIREX-AQ field campaign
(https://csl.noaa.gov/projects/firex-aq/) employed a suite of satellites, aircraft, vehicles and
ground site platforms aimed to observe, analyze, and characterize air pollutants emitted from
wildfire sources over the CONUS (Ye et al., 2021).  The FIREX-AQ airborne measurements



provide a three-dimensional dataset from various meteorological, gas, and aerosol instruments
that is used to verify the GFS-CMAQ and WRF-CMAQ model performance, while elucidating
reasons for any model differences.  Here the focus of the FIREX-AQ model comparison and
verification is against observations taken primarily from the NASA DC-8 aircraft, which include
meteorological variables, gaseous and aerosol concentrations, and aerosol optical properties. The
majority of the FIREX-AQ flights were over the western United States, and sampled within
environments that both were *and* were not (see section 4.1) influenced by wildfire emissions
(https://daac.ornl.gov/MASTER/guides/MASTER_FIREX_AQ_JulySept_2019.html). During a
cluster of major wildfire events (see Section 4.2), the DC-8 sampled both near-source and aged
smoke plumes between 02–08 August, 2019 (i.e., the Williams Flats, Snow Creek, and Horsefly
Fires) across the states of Idaho, Washington, and Montana.

### 4.1 Comparison of the July 22 non-wildfire event over the central California Valley

On 22 July, the DC-8 aircraft flew from California to Boise, Idaho, while maintaining a
relatively low-altitude (<1 km) above sea level (ASL) over the California Central Valley (Figure
7). This flight was not impacted by any major wildfire event, and was mainly controlled by
anthropogenic emissions and local meteorological conditions. Figure 7 shows that the GFSv16
and WRF models had similar meteorological temperature and humidity predictions, and that both
models have dry and warm biases over the Central Valley at lower altitudes (Figures 7d–7e)
(Yun et al., 2020).  GFS's horizontal wind speeds tended to have stronger variability than WRF
(Figure 7b), especially in high altitudes. For wind direction, WRF showed a better prediction
than GFS around 20 and 24 UTC (Figure 7c).
Both GFS-CMAQ and WRF-CMAQ underestimate the vertical wind (W) variability by at least
one order of magnitude, and WRF-CMAQ has weaker W variability than that of GFS-CMAQ,
especially in high altitudes (Figure 7f). The model vertical velocities are not from the GFS or
WRF model, but rather they are re-diagnosed in CMAQ to conserve mass  (Otte and Pleim,
2010), and thus represent the whole layer's vertical movement across the 12 km by 12 km grid
cell. With its flight speed around 80 to 240 m/s, the DC-8 aircraft's one-minute average
sampling frequency results in an approximate 4.8 to 14 km horizontal scale, respectively, which
is comparable to the 12 km CMAQ model resolution. The aircraft observations, however, include
turbulence effects during its one-minute averages, which may not be temporally resolved by
CMAQ at this resolution. Thus, both the GFS-CMAQ and WRF-CMAQ model vertical
velocities are much lower and have almost no correlation with the aircraft observations.
Although both GFS-CMAQ and WRF-CMAQ have reasonable comparisons for most
meteorological variables, including the horizontal winds, it continues to be a challenge to
compare them with the observed vertical velocities. Thus to further elucidate the model vs.
observation differences in vertical motions, Figure 8 shows a  curtain plot of vertical velocities
along the flight path from the two models. Since WRF-CMAQ remains in a native grid, its wind




fields tend to be more balanced and have lower variability compared to the interpolated GFS-
CMAQ wind fields. The stronger variability in W for GFS-CMAQ represents CMAQ's effort to
counteract mass inconsistency effects from the interpolated horizontal wind fields (Byun,
1999b).
GFS-CMAQ and WRF-CMAQ overall yield similar results for specific chemical species during
this DC-8 flight (Figure 9). Both models underestimate CO, $O_3$ and ethane ($C_2H_6$) concentrations
over the lower altitudes in the California Central Valley. Over the same flight segment, they had
better NOx (NO + $NO_2$) and ethene ($C_2H_4$) predictions, implying that the emissions of these two
species have better accuracy than those of CO and ethane. Figure 9f shows that the two models
also underestimate NOz (NOy–NOx), or the oxidized nitrogen species besides NOx, indicating
that the photochemical ozone production may also be underestimated. NOz is a good indicator of
the ozone photochemical formation (Sillman et al., 1997), where the $O_3$/NOz ratio represents the
ozone photochemical efficiency per NOx oxidation products. Thus, NOz and $O_3$ are typically
highly correlated over regions with active photochemical production. The $O_3$ and NOz
underestimations are likely due to the underestimation of CO and some hydrocarbons, such as
ethane, as they are precursors of $O_3$.
The two models show slight differences in peak values of CO, ethene, and NOx around 23:30
UTC, where the GFS-CMAQ predicted concentrations are slightly higher and closer to
observations (Figure 9). These differences are due to their PBL predictions (both from the
corresponding meteorological model outputs), where GFS-CMAQ has a lower PBL height and
weaker emission vertical dilution compared to WRF-CMAQ (Figure 8). GFS-CMAQ tends to
underpredict $O_3$ more (Figure 9b), however, due to its higher NOx titration.  This implies that the
effects of the transport and non-local transformation of $O_3$ could be stronger than that of local
precursor emissions. WRF-CMAQ has higher NOz (Figure 9f), but lower NOx compared to
GFS-CMAQ due to the time lag of $O_3$ and NOz photochemical formation. Consequently, the
peak $O_3$ values may not be well correlated with the emitted precursors, such as NOx and volatile
organic compounds (VOCs). Furthermore, the modeled peak $C_2H_6$ and $C_2H_4$ concentrations do
not occur at the same time around 23:30 UTC, while observations indicate that these two species
should be highly correlated in this region. This model mismatch implies that the VOC speciation
factors for a certain area or emission sector need to be improved over Southern California.
**4.2 Comparison of the 6 August wildfire events over the U.S. Northwest**
On 06 August, the DC-8 observed a cluster of three wildfires: the Williams Flats Fire (47.98 °N,
118.624 °W, 80 km to the northwest of Spokane, Washington), Snow Creek Fire (47.703°N,
113.4°W, 32 km northeast of Condon, Montana), and Horsefly Fire (46.963 °N, 112.441°W, 24
km east of Lincoln, Montana). Figure 10a shows the flight path on that date, where the DC-8
aircraft departed from Boise, ID,  flew over the Williams Flats Fire region, then flew to Montana
to sample the Snow Creek and Horsefly Fires (i.e., Montana Fires), and finally returned to the





Boise base. The aircraft flew below 8 km for most flight segments near the fire plumes. Figure
S1 shows the corresponding GOES-16 satellite true color image, where these 06 August fires and
associated smoke plumes are visible and can be distinguished from the cloud bands to the south
that move northward later that day (Figure S2).  The Williams Flats Fire was ignited by
lightning, and was the largest fire event sampled during the FIREX-AQ campaign burning from
about 02–08, August, 2019.
Both models significantly underpredicted CO (Figure 10c), submicron organic aerosol (Figure
10e) and aerosol optical extinction coefficient (AOE) (Figure 10f), which suggests an issue with
the GBBEPx gas and aerosol emissions. The models performed well for $NO_2$ during the
Williams Flats and Montana Fires Fire below 6 km ASL, but there were prominent
underestimations for the high-altitude flight segments (Figure 10d). This indicates that the
background $NO_2$ was underestimated, or the models had insufficient inject height for fire plume
rise (both based on Briggs, 1965). WRF-CMAQ predicted higher $O_3$ values than the GFS-
CMAQ, which overall agreed better with observations for the Williams Flats Fire (Figure 10b).
However, for the Montana Fires (~ 23–24 UTC), WRF-CMAQ has higher $O_3$ biases and GFS-
CMAQ yields better results. The difference in $O_3$ is largely driven by the background
concentration difference between the two models, where WRF-CMAQ tends to have higher
domain-wide $O_3$ than GFS-CMAQ due to the meteorological effects discussed in Sections 3.
Figure S3 shows the spatial overlay comparison of vertically averaged GFS-CMAQ predictions
at 21 UTC and the DC-8 flight observations for the altitude 1–3 km above ground level (AGL),
on 6 August, 2019. The peak $NO_2$ observation around 118.5°W, 48°N indicates the general
location of the Williams Flats fire. The GBBEPx emission and GFS-CMAQ prediction showed
shifted peak-value locations driven by the westerly modeled winds. For this flight, the GBBEPx
had stronger NOx fire emission over two Montana locations than that over Williams Flats. The
model overpredicts the column averaged $NO_2$ concentrations, especially over the Montana fires,
which can not be reflected by the point-by-point $NO_2$ comparison result in Figure 10d. For this
flight, the mean GFS-CMAQ $NO_2$ along the flight path for 1–3km AGL is about 0.125 ppbv
compared to the observed mean $NO_2$ of 0.169 ppbv, and the model indeed showed $NO_2$
underprediction along the flight path. However, in this case, the flight path missed some
locations where modeled peak $NO_2$ values existed or the modeled transport misplaced the
plumes, especially over the Montana fires leading to this inconsistency. For ozone comparison
(Figure S3b), this inconsistency could also exist, though not as significant as for the high-
gradient $NO_2$ concentrations. In the GFS-CMAQ prediction, the high ozone concentrations are
almost co-located with high $NO_2$ concentration (Figure S3b), but the observation did not show
this feature. Instead, some high-$O_3$ flight segments had relatively low $NO_2$, such as those circled
in the black rectangle box of Figure S3b. The observed NOx titration was not able to be
produced by the 12 km models. Wang et al (2021) used a 100m horizontal resolution large eddy
simulation and demonstrated the capability of using such techniques to capture some high-



resolution fire plume features and associated chemical behavior. While such high resolution
techniques are not currently feasible for the operational NAQFC, they demonstrate the limitation
of using regional scale (12×12 km) models to capture such fine scale features of plume behavior.
GFS-CMAQ has higher wildfire-related CO, $NO_2$, OA and AOE values that are closer to
observations than WRF-CMAQ for the Montana Fires between 23–24 UTC at flight altitudes of
~ 4–5 km (Figure 10c–10f). Since these two models use the same GBBEPx emissions and
wildfire plume rise algorithm (Briggs, 1965), the differences should be due to other reasons. To
help explain these model differences, Figure 11a and 11b show the Differential Absorption High
Spectral Resolution Lidar (DIAL-HSRL) retrieved aerosol backscatter coefficients (ABC)
aboard the DC-8 aircraft without and with cloud screen, respectively. It shows that the major fire
plumes of the William Flats Fire were below 4 km (~ 19–22 UTC) , but the Montana Fires (~23–
24 UTC) extended from the surface up to 6 km, with some detached plumes reaching 10 km. The
model predicted AOEs have an overall similar pattern, with major plumes below 4 km for the
Williams Flats Fire (Figures 11c and 11d). Over the Montana Fires, the GFS-CMAQ predicts
slightly higher PBL, thus allowing for the fire plume to reach a higher height near the DC-8
cruising altitude.  In contrast, the WRF-CMAQ wildfire plumes are slightly lower than the
aircraft flight path around 23–24 UTC, which leads to underprediction in the fire emitted species
(Figure 11d).
An interesting feature in the DIAL observations is the detached plume from 8 km to 10 km
altitude (Figure 11a), where some cirrus clouds existed, and the DIAL retrieval could not
distinguish whether they are pure clouds or clouds mixed with elevated aerosols above 8km. The
cloud screen product (Figure 11b) mainly showed the enhanced aerosols below 7km and some
scattered signals near the high cloud edges. Cloud mixing with aerosols was usual for fire
induced clouds, or pyrocumulonimbus (Peterson et al., 2021). Although in this event, the middle-
size fires did not show evident of inducing the high-altitude clouds, the indicator of mixed clouds
and aerosols in high altitudes still existed: both in-situ measured OA (Figures 10e) and AOE
(Figure 10f, 11c, 11d) showed the enhanced aerosols around 25 UTC above 8km. This elevated
plume was generally captured by the GFS-CMAQ simulation, while underestimating its strength
(Figure 11c); however, this feature was completely missed in WRF-CMAQ (Figure 11d).
Considering the altitude range of the detached plume, the major model disparities are likely due
to model convection differences in the free troposphere. To further investigate this impact,
Figures 11e and 11f show curtain plots of RH predicted by the two models. GFS-CMAQ yields
higher RH at such altitudes (10 km) compared to WRF-CMAQ around 23–24 UTC, indicating
that the GFS-CMAQ has stronger convection. The CMAQ model uses inputted meteorology to
diagnose convection activity and drive its ACM2 convection scheme. This convective activity is
apparent in GOES-16 satellite images (Figure S2), as more fractional clouds appeared ahead of
the northward moving frontal band. Both the GFSv16 and WRF models used here *do not*
consider the fire heat feedback effect, and thus their predicted convection and clouds are only





driven by the synoptic weather conditions. If such synoptic-to-mesoscale weather models
consider wildfire heat feedback effects, their predictions may result in stronger convection and
help correct underpredictions in PBL heights.
**4.3 Statistical Results of Model Performances for FIREX-AQ**
4.3.1 *Meteorological Statistics*
During the FIREX-AQ field campaign, the DC-8 aircraft performed more than 20 flights over
CONUS with detailed observations of various chemical compounds. Tables 2 and 3 show the
statistical results of mean bias (MB), normalized mean bias (NMB), root mean square error
(RMSE), correlation coefficient (R), and linear regression/slopes for the two models'
performance over the western U.S. (west of 110°W) only at low altitudes (<3km ASL) for both
non-fire and fire flight segments. Most of these flights departed from Boise, ID, except the 22
July flight that flew from California to Idaho. As a result, they mainly flew over Idaho and its
surrounding regions. The GFS tends to have slightly higher wind speed with positive MB, while
WRF has a small negative wind speed bias. Most of the DC-8 flights are during the daytime, and
the GFS has a higher daytime wind speed than WRF at low altitudes. The GFS and WRF have
very similar temperature predictions. For the RH, the GFS predictions were slightly dryer than
those of WRF, especially for non-fire events. The meteorological models do not consider
wildfire heat effects, and thus may have (in part) led to slightly warm MB for the non-fire events
(Table 2) and slightly cool MB for the fire events (Table 3). Because both the GFSv16 and WRF
models have similar MB shifts from an average temperature overprediction (Table 2; non-fire
events) to an underprediction (Table 3; wildfire events), we can estimate that the fire effects
made roughly a 1–2 Kelvin temperature enhancement to the background along the DC-8 flight
paths below 3 km. This estimate assumes that the model temperature biases are generally
representative of the western U.S. (west of 110°W), and are independent of the averaged flight
segments that have different locations and periods in Table 2 and Table 3. Correspondingly, the
air masses are dryer in the sampled wildfire plumes, as shown by the large reduction in the RH
underpredictions (i.e., negative MBs) from Table 2 to Table 3.
4.3.2 *Chemical Statistics During Non-Fire Events*
For most chemical species, the two models also have similar performance, indicating that the
emissions and chemistry are major driving forces. For non-fire events, both models overpredict
NOx, $HNO_3$, toluene, EC, and ammonium ($NH_4^+$), but underestimate PAN, benzene, $C_2H_2$, $SO_2$,
and submicron sulfate and organic aerosols (OA) (Table 2). The $SO_2$ and submicron sulfate
underprediction may be impacted by underestimated NEIC2016v1 $SO_2$ emissions over the
western U.S. Since point sources, including power plant emissions, are the $SO_2$ sources, this
comparison implies that the point sources for 2019 events have large uncertainties.





Although the models agree well with NOy observations, they disproportionately underestimate
NOz as shown by the regression slopes and MBs. One of the important NOz species is PAN, and
both models underestimate PAN during the non-fire events (Table 2).  PAN's carbonyl
precursors include acetaldehyde ($CH_3CHO$) (44% of the global source), methylglyoxal (30%),
acetone (7%), and a suite of other isoprene and terpene oxidation products (19%) (Fischer et al.,
2014). $CH_3CHO$ and acetone are also underestimated (Table 2), and help explain PAN's
underestimation. For the oxidized hydrocarbons, like aldehydes (HCHO, $CH_3CHO$), their main
atmospheric sources are the oxidation of highly reactive VOCs, including alkanes, alkenes, and
aromatics, instead of direct emissions (Parrish et al., 2012). So, the underestimation of HCHO
and CH3CHO are associated with the underestimation of their precursor hydrocarbons, including
anthropogenic and biogenic VOCs. Our other comparison indicated that BEIS tends to
underpredict biogenic emission over the U.S. West, e.g. isoprene in Table 2. In this comparison,
most anthropogenic hydrocarbons are disproportionately underestimated, except toluene,
implying the VOC speciation issue in the NEIC2016v1 anthropogenic emissions (Table 2).
Previous work had discovered that a model overprediction in toluene was also related to the
toluene speciation in the NEI emission inventory (Lu et al., 2020).
Submicron ammonium ($NH_4^+$) and the nitrate ion are also underestimated by both models during
non-fire events (Table 2), suggesting there are $NH_3$ underestimates due to either insufficient $NH_3$
emissions or exaggerated $NH_3$ removal processes. There are, however, overpredictions in the
intermediate species nitric acid ($HNO_3$). It implies that the $HNO_3$ accumulates in the atmosphere
because the modeled NOz pathways toward the nitrate ion and organic nitrate aerosol products
are reduced due to their other precursor ($NH_3$ and VOCs) underestimation.
There are underestimations in the VOC and CO concentrations, which contributes to the ozone
underestimation during non-fire events (Table 2). These non-fire comparisons also highlight that
both models have similar biases due to similar meteorology (Section 4.3.1), and the use of the
same anthropogenic emissions (NEIC2016v1), BEIS biogenic emission and chemical
models/mechanisms (i.e CMAQv5.3.1).  The differences in the two models' bias, error, and
correlation/slope are much smaller than their individual magnitudes.
4.3.3 *Chemical Statistics During Fire Events*
The WRF-CMAQ and GFS-CMAQ models significantly underestimate CO, VOC, HONO, and
OA for fire events at low altitudes (< 3 km) over the western U.S. (Table 3). In conjunction with
underestimated GBBEPx emissions during these wildfire events, other possible causes for the
average statistical underprediction are the CMAQ model's 12 km horizontal resolution and the
flight sampling coverage. Most of the fires that are averaged in the statistics, such as the Horsefly
(5.5 $km^2$ burning area) and Snow Creek Fires (7.3 $km^2$ burning area), are at a much finer scale
than the model grid. Only the largest Williams Flats Fire, with a total burning area of 180 $km^2$
(Ye et al., 2021), had a comparable horizontal scale to the model resolution.




The DC-8 aircraft had many flight segments near wildfire sources during the fire events in Table
3, and thus dilution of the emissions due to the relatively coarse model resolution may lead to
underestimations in the predicted slope for most wildfire emitted pollutants, such as CO and OA
(Table 3). The $O_3$ concentrations are also underestimated; however, the $O_3$ underpredictions are
reduced from the non-fire (Table 2) to fire events (Table 3). Abundant amounts of wildfire
emitted NOx can titrate ozone near the fire source region, and the models likely underestimate
these titration effects due to the 12 km model resolution. Thus, the models cannot capture the
strong spatial $O_3$ variability that is observed due to both reduction near source regions and
enhancement in downstream areas. Again, for this fire event comparison, both models showed
similar behavior and their differences were relatively smaller compared to the overall model
biases.
## 5. Summary and Discussion
The operational NOAA/NWS National Air Quality Forecasting Capability (NAQFC) recently
underwent a major upgrade on July 20, 2021. The advanced NAQFC includes the recent
Community Multi-scale Air Quality (CMAQ) model version 5.3.1 with CB6 (carbon bond
version 6)-Aero7 (version 7 of the aerosol module) chemical mechanism, and is driven by the
latest operational Finite Volume Cubed-Sphere (FV3)-Global Forecast System, version 16
(GFSv16) (Campbell et al., 2022). Here we analyze the impacts of the driving meteorological
models on CMAQ model performance with the new GFSv16 interpolation-based meteorology
versus the commonly-used native-grid Weather Research and Forecasting (WRF) model version
4.0.3 meteorology. The meteorological and chemical analysis includes both 2D ground-based
and 3D aircraft measurements during the summer 2019, which encompasses the joint NOAA-
NASA Fire Influence on Regional to Global Environments and Air Quality (FIREX-AQ)
campaign. As CMAQ has existing mass conservation via adjustments of the contravariant
vertical velocity (Otte and Pleim, 2010), the NACC interpolated GFSv16 wind field can be well
handled in CMAQ (i.e., GFS-CMAQ).
The different NWS/NOAA operational GFS and commonly chosen WRF physics schemes
employed in this study (Table 1) clearly have impacts on temperature, wind fields (both
horizontal/advection and vertical/convection), PBL heights, and the corresponding CMAQ
model predictions. During this study period over the U.S. West, both models showed moisture
dry bias and temperature warm bias in low altitudes, which could be due to the issue mentioned
by Qian et al (2020) and impacts from soil moisture deficits on surface fluxes in both models.
Due to their different physics, GFS had stronger diurnal variation of PBL height, lower at night
and higher during daytime over the U.S. West and Northeast. The differences in the GFS and
WRF physics have a larger impact than the meteorology driver methodologies (i.e., interpolation
vs. native) on the models' meteorological and air quality predictions, even despite using FDDA
to nudge WRF simulation toward the GFSv16 data. While FDDA nudging was used here in





WRF to avoid growing errors across a continuous 1-month simulation, we note that if it is turned
off, the differences between GFSv16 and WRF predictions would have been even greater. This
would further substantiate the dominance of using different model physics compared to using
different meteorological-driver methodologies (i.e., native vs. interpolation) and their impacts on
CMAQ model predictions. Overall, the results of this study further corroborate the use of the
GFSv16 data and NACC interpolation-based methods (Campbell et al., 2022) for regional
CMAQ model applications in the scientific community.
Over CONUS, GFS-CMAQ demonstrated lower mean surface ozone (by about 3 ppb) and
PM2.5 (by about 1 $\mu g/m^3$) than WRF-CMAQ in August 2019 (section 3). In the western U.S.,
the GFS has a stronger diurnal variability in PBL height and a better performance in nighttime
10-m wind speeds compared to WRF. The nighttime difference between these two models tends
to be more significant than the corresponding daytime difference. Their difference is also
impacted by both vertical/convective (mainly daytime) and upstream advective transport
differences in GFS-CMAQ and WRF-CMAQ, which somewhat confounds the impact of
different meteorological physics on chemical predictions from region to region. This transport
effect is more significant on PM2.5 than that on $O_3$, as $O_3$ has a shorter lifetime and is more
sensitive to local emissions in summer. In this study, neither GFS-CMAQ nor WRF-CMAQ
show overwhelming performance advantage over the other, similar to the NMM-CMAQ and
ARW-CMAQ comparison in Yu et al. (2012a, 2012b).
GFS-CMAQ and WRF-CMAQ demonstrated rather similar performance for major chemical
variables during both FIREX-AQ non-fire (Table 2) and fire events (Table 3). In most FIREX-
AQ events, both GFS-CMAQ and WRF-CMAQ showed similar biases, indicating that other
factors, including emissions, model resolution and chemistry etc. could be more important for the
model predictions compared to the meteorological differences. The aircraft data comparison
reveals many common issues in both model systems. One critical issue is whether the flight
sampling coverage is comparable to the 12 km model resolution, especially for high-gradient fire
emission, e.g. the case of 06 August flight (Figure S3). The observation representation issue also
exists in other places, such as near-surface meteorological comparison between AIRNow stations
and METAR stations. Emission is the driving force for atmospheric composition concentrations.
The comprehensive aircraft measurements help verify that the anthropogenic NEIC2016v1
inventory is overall reasonable, except for $SO_2$, $NH_3$ and certain hydrocarbons. The wildfire
emission has bigger uncertainties, including the emission intensities, specification and plume
rise, shown by the both models' results.
The NACC interpolation method is advantageous as it enables using the original meteorological
driver directly via interpolation, and avoids running another model such as WRF as a downscaler
for regional CMAQ applications. It is also faster, and more consistent with the original
meteorological driver. These aspects can simultaneously benefit real-time forecasting and



retrospective air quality applications in the scientific community. NACC can also adapt to
quickly use any regional domain globally, and may also use other global meteorological data
including reanalysis products.  This helps mitigate the confounding factors of using different
model configurations across the myriad of WRF physics options, while alleviating the difficulty
in understanding their impacts on air quality predictions. The operational GFSv16 and associated
reanalysis products are well vetted and evaluated across different global agencies and
laboratories, and thus are well suited for regional CMAQ applications using NACC. In fact, there
is an ongoing project at NOAA to migrate both the GFSv16 data and NACC software to the
Amazon Web Services (AWS) Cloud platform to provide a streamlined product for the user to
generate the model-ready meteorological data for any regional CMAQ application globally.
Finally, we note that the current operational GFSv16 has enough meteorological variables to
drive CMAQ with other supplied data (fractional landuse, LAI etc), and its C768 grid has
horizontal resolution from 10.21 km to 14.44 km, which is close to the NAQFC's 12 km
horizontal resolution. However, some commonly available global meteorological data, such as
NCEP or ECMWF reanalysis data, may not have all meteorological variables needed by CMAQ,
and have relatively coarse model resolutions. In this case, the WRF downscaling may become
the only available method to drive a finer scale CMAQ model application. WRF data generated
by different physics may be good for a finer scale CMAQ simulation; however NACC
developments are underway to also process/interpolate higher resolution FV3-based Limited
Area Models (LAMs) for direct application to CMAQ. All the physics schemes were developed
according to certain regions and meteorological conditions. We again stress, however, that the
downscaled WRF physics may significantly alter the original meteorological fields even with the
FDDA nudging. As shown in this study, GFS and WRF had mixed performance for driving
CMAQ.
**Code and Data Availability**.
The FIREX-AQ field campaign data used in this study is in [https://www-air.larc.nasa.gov/cgi-](https://www-air.larc.nasa.gov/cgi-bin/ArcView/firexaq)
[bin/ArcView/firexaq](https://www-air.larc.nasa.gov/cgi-bin/ArcView/firexaq) (last access, 16 May 2022). The NACC code used in this study is publicly
available at https://doi.org/10.5281/zenodo.5507489 and via GitHub at https://github.com/noaa-
oar-arl/NACC.git (last access: 5 April 2022). The modified CMAQv5.3.1 for GFS-CMAQ is
available at https://doi.org/10.5281/zenodo.5507511 and via GitHub at https://github.com/noaa-
oar-arl/NAQFC (last access: 5 April 2022).
**Author contributions.**
YT contributed to the project conceptualization, model run, software, data analysis, visualization,
investigation, and writing of the original draft. PCC contributed to software, the model run, data
analysis, investigation and draft revision. DT and XZ contributed to wildfire emissions data. BB
contributed to software and funding acquisition. FY, JH and HH provided the GFS model data.
LP provided the global aerosol model for the lateral boundary condition. PL, RS, AS, JF, IS, JT-



D, YJ contributed to project supervision, project administration, and funding acquisition. MY, IB, JF, TR, DB, JS, J-LJ, JC, GD, RM, JH, GH, AR and JD contributed to the FIREX-AQ aircraft data.

**Competing interests.**

The contact author has declared that neither they nor their co-authors have any competing interests.

## Acknowledgements

This research was funded by NOAA's National Air Quality Forecasting Capability (NAQFC) in the National Weather Service Office of Science and Technology Integration (NWS/OSTI).

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





1   Table 1. The two meteorological datasets used in this study

| Model Settings | FV3-GFSv16/NACC | WRF-ARW/MCIP |
|---|---|---|
| Domain | Global C768L127 (~ 13 km horizontal resolution in 6 cubic spherical tiles , 127 vertical layers up to 80km), interpolated to the 12km CONUS domain with 35-layers up to about 14km (60hPa) | 12km CONUS 35 vertical layers up to 100hPa |
| Dynamic core | Finite Volume 3, non-hydrostatic (Putman and Lin, 2007) | WRF-ARW dynamic in hybrid vertical coordinate (Skamarock et al., 2021) |
| Initial condition | FV3-GFSv16 analysis (GDAS) using the local ensemble Kalman filter (LETKF) (Ott et al., 2004) with 4-dimensional incremental analysis update (4D-IAU) | FV3-GFSv16 analysis (GDAS) |
| Lateral Boundary Condition | N/A | FV3-GFSv16 analysis (GDAS) |
| Cloud Microphysics | GFDL six-category cloud microphysics scheme (Lin et al., 1983; Lord et al., 1984; Krueger et al., 1995; Chen and Lin, 2011; Chen and Lin, 2013) | Morrison 2-moment scheme (Morrison et al., 2009) |
| PBL Physics Scheme | Scale-aware (sa) turbulent kinetic energy (TKE) -based moist eddy-diffusivity mass-flux (EDMF) (sa-TKE-EDMF) (Han and Bretherton,2019) | Yonsei University Scheme (Hong et al., 2006) |
| Shallow/Deep Cumulus Parameterization | SAS Scheme (Han et al. 2011; 2017) | Kain Fritsch multiscale (Kain, 2004) |
| Shortwave and Longwave Radiation | RRTMG (Mlawer et al. 1997; Clough et al. 2005; Iacono et al. 2008) | RRTMG (Iacono et al. 2008). |
| Land Surface Model | Noah Land Surface Model (Chen and Dudhia 2001; Ek et al. 2003;Tewari et al. 2004) | Noah (Tewari et al., 2004) |
| Surface Layer | Monin-Obukhov (Monin-Obukhov 1954; Grell et al. 1994; Jimenez et al. 2012) | Revised MM5 Scheme (Jimenez et al., 2012) |
| Other treatment | | FDDA nudging is enabled for temperature and specific humidity whole domain, and for wind components (U, V) outside the PBL. |



| Variables | Obs Mean | GFS-CMAQ | | | | | WRF-CMAQ | | | | |
|---|---|---|---|---|---|---|---|---|---|---|---|
| | | MB | NMB | RMSE | R | Slope | MB | NMB | RMSE | R | Slope |
| Temperature (K) | 295 | 0.979 | 0.332 | 2.04 | 0.988 | 1.13 | 1.16 | 0.393 | 2.28 | 0.989 | 1.17 |
| RH (%) | 35.6 | -7.3 | -20.5 | 11.8 | 0.781 | 0.717 | -6.05 | -17 | 12.6 | 0.677 | 0.598 |
| Wind Speed (m/s) | 4.81 | 0.758 | 15.8 | 3.25 | 0.432 | 0.473 | -1.11 | -23.1 | 2.4 | 0.666 | 0.524 |
| $O_3$ (ppbv) | 57.9 | -10.7 | -18.5 | 15 | 0.651 | 0.34 | -10.4 | -17.9 | 14.1 | 0.717 | 0.413 |
| CO (ppbv) | 134 | -37.6 | -28 | 53.2 | 0.654 | 0.573 | -37.1 | -27.7 | 52.9 | 0.652 | 0.572 |
| NOx (ppbv) | 1.11 | 0.507 | 45.6 | 2.9 | 0.704 | 1.15 | 0.345 | 31.1 | 2.86 | 0.695 | 1.12 |
| NOy (ppbv) | 2.56 | -0.0418 | -1.63 | 3.07 | 0.743 | 0.892 | 0.055 | 2.15 | 3.14 | 0.724 | 0.86 |
| NOz (ppbv) | 1.63 | -0.465 | -28.6 | 1.17 | 0.782 | 0.553 | -0.125 | -7.66 | 1.08 | 0.788 | 0.721 |
| HONO (ppbv) | 0.00432 | 0.012 | 279 | 0.0438 | 0.379 | 0.444 | 0.0134 | 311 | 0.0487 | 0.358 | 0.48 |
| $HNO_3$ (ppbv) | 0.291 | 0.154 | 53.1 | 0.421 | 0.683 | 1.34 | 0.337 | 116 | 0.65 | 0.708 | 1.89 |
| PAN (ppbv) | 0.399 | -0.251 | -63 | 0.416 | 0.675 | 0.221 | -0.222 | -55.6 | 0.386 | 0.681 | 0.284 |
| $NH_3$ (ppbv) | 3.55 | -0.801 | -22.6 | 5.26 | 0.0481 | 0.038 | -1.58 | -44.5 | 4.37 | 0.304 | 0.155 |
| $C_2H_4$ (ppbv) | 0.121 | 0.0582 | 48.1 | 0.189 | 0.702 | 0.869 | 0.0385 | 31.9 | 0.187 | 0.682 | 0.836 |
| $C_2H_2$ (ppbv) | 0.146 | -0.0734 | -50.3 | 0.137 | 0.784 | 0.496 | -0.0696 | -47.7 | 0.137 | 0.771 | 0.494 |
| $SO_2$ (ppbv) | 0.342 | -0.235 | -68.8 | 0.567 | 0.0238 | 0.00835 | -0.221 | -64.5 | 0.568 | $-1.26\times10^{-3}$ | -0.00047 |
| Acetone (ppbv) | 2.74 | -2.28 | -83.1 | 2.45 | 0.686 | 0.192 | -2.2 | -80.4 | 2.38 | 0.668 | 0.199 |
| HCHO (ppbv) | 2.1 | -0.972 | -46.4 | 1.26 | 0.559 | 0.447 | -0.909 | -43.4 | 1.25 | 0.513 | 0.442 |
| $CH_3CHO$ (ppbv) | 0.736 | -0.326 | -44.2 | 0.538 | 0.647 | 0.386 | -0.349 | -47.4 | 0.554 | 0.643 | 0.38 |
| Benzene (ppbv) | 0.0449 | -0.0193 | -43 | 0.057 | 0.398 | 0.385 | -0.0191 | -42.6 | 0.0564 | 0.397 | 0.375 |
| Toluene (ppbv) | 0.039 | 0.0409 | 105 | 0.153 | 0.759 | 1.74 | 0.0352 | 90.1 | 0.14 | 0.762 | 1.63 |
| Isoprene (ppbv) | 0.073 | 0.0361 | 49.4 | 0.174 | 0.6 | 0.838 | 0.00661 | 9.06 | 0.145 | 0.648 | 0.797 |
| EC ($\mu g/std\ m^3$) | 0.108 | 0.191 | 177 | 0.572 | 0.518 | 2.09 | 0.228 | 211 | 0.609 | 0.455 | 1.88 |
| OA ($\mu g/std\ m^3$) | 10.9 | -7.15 | -65.7 | 9.72 | 0.565 | 0.263 | -6.48 | -59.5 | 9.45 | 0.495 | 0.243 |
| Sulfate ($\mu g/std\ m^3$) | 1.31 | -0.781 | -59.7 | 1.11 | 0.0856 | 0.0188 | -0.773 | -59 | 1.11 | 0.0322 | 0.00677 |
| $NH_4^+$ ($\mu g/std\ m^3$) | 0.745 | -0.615 | -82.5 | 0.805 | 0.416 | 0.103 | -0.596 | -79.9 | 0.778 | 0.509 | 0.145 |
| Nitrate ($\mu g/std\ m^3$) | 1.22 | -1.08 | -88.1 | 1.49 | 0.562 | 0.229 | -1.04 | -85.3 | 1.45 | 0.57 | 0.279 |
| AOE (/Mm) | 54.5 | -29.3 | -53.8 | 47 | 0.593 | 0.227 | -27.4 | -50.2 | 45.9 | 0.588 | 0.227 |

Table 2. Statistics of the two models compared to the observation for DC-8 flight segments with non-fire events below 3km (ASL) over west of -100°W. All aerosols are in submicron. The normalized mean bias (NMB) is in unit %.



| Variables | Obs Mean | GFS-CMAQ | | | | | WRF-CMAQ | | | | |
|---|---|---|---|---|---|---|---|---|---|---|---|
| | | MB | NMB | RMSE | R | Slope | MB | NMB | RMSE | R | Slope |
| Temperature (K) | 287 | -0.389 | -0.135 | 0.702 | 0.995 | 1.01 | -0.688 | -0.24 | 0.863 | 0.997 | 1.04 |
| RH (%) | 27.8 | -0.761 | -2.74 | 7.84 | 0.712 | 0.553 | 4.3 | 15.5 | 11.1 | 0.556 | 0.534 |
| Wind Speed (m/s) | 5.42 | 0.766 | 14.1 | 2.16 | 0.612 | 0.616 | -0.811 | -15 | 2.12 | 0.604 | 0.556 |
| $O_3$ (ppbv) | 55.7 | -6.61 | -11.9 | 11.8 | 0.587 | 0.262 | -7.01 | -12.6 | 11.5 | 0.653 | 0.346 |
| CO (ppbv) | 486 | -377 | -77.6 | 873 | 0.596 | 0.0347 | -383 | -78.8 | 883 | 0.442 | 0.0242 |
| NOx (ppbv) | 2.63 | 0.06 | 2.28 | 6.41 | 0.465 | 0.231 | -0.619 | -23.5 | 7.02 | 0.31 | 0.153 |
| NOy (ppbv) | 7.32 | -4.19 | -57.3 | 13.3 | 0.507 | 0.123 | -4.66 | -63.7 | 14.2 | 0.31 | 0.073 |
| NOz (ppbv) | 5.7 | -4.8 | -84.3 | 10.2 | -0.189 | -0.0106 | -4.68 | -82 | 10.2 | -0.204 | -0.0121 |
| HONO (ppbv) | 0.283 | -0.274 | -96.8 | 1.18 | 0.355 | 0.0043 | -0.274 | -96.8 | 1.18 | 0.291 | 0.00457 |
| $HNO_3$ (ppbv) | 0.148 | 0.148 | 99.7 | 0.256 | 0.532 | 1.07 | 0.179 | 121 | 0.28 | 0.402 | 0.768 |
| PAN (ppbv) | 0.971 | -0.793 | -81.7 | 1.63 | 0.27 | 0.0195 | -0.765 | -78.8 | 1.61 | 0.279 | 0.026 |
| $NH_3$ (ppbv) | 17.7 | -12.3 | -69.3 | 28.3 | 0.379 | 0.0654 | -13.7 | -77.4 | 29.6 | 0.232 | 0.0386 |
| $C_2H_4$ (ppbv) | 4.5 | -4.34 | -96.3 | 10.2 | 0.421 | 0.00498 | -4.36 | -96.8 | 10.2 | 0.14 | 0.0018 |
| $C_2H_2$ (ppbv) | 1.04 | -1.01 | -96.9 | 2.08 | 0.534 | 0.00866 | -1.01 | -97 | 2.09 | 0.363 | 0.00623 |
| $SO_2$ (ppbv) | 0.699 | -0.322 | -46.1 | 1.38 | 0.589 | 0.198 | -0.392 | -56.1 | 1.5 | 0.429 | 0.132 |
| Acetone (ppbv) | 3.54 | -3.2 | -90.3 | 4.56 | 0.13 | 0.00862 | -3.18 | -89.7 | 4.55 | 0.135 | 0.0112 |
| HCHO (ppbv) | 8.17 | -7.13 | -87.3 | 17.8 | 0.232 | 0.0062 | -7.19 | -88 | 17.8 | 0.119 | 0.00303 |
| $CH_3CHO$ (ppbv) | 3.65 | -3.18 | -87.4 | 9.13 | 0.186 | 0.00547 | -3.21 | -88 | 9.2 | -0.027 | -0.00097 |
| Benzene (ppbv) | 0.683 | -0.67 | -98.1 | 1.84 | 0.54 | 0.00432 | -0.672 | -98.3 | 1.84 | 0.367 | 0.00275 |
| Toluene (ppbv) | 0.451 | -0.436 | -96.6 | 1.36 | 0.402 | 0.00491 | -0.438 | -97 | 1.36 | 0.195 | 0.00245 |
| Isoprene (ppbv) | 0.095 | $-7.9 \times 10^{-3}$ | -8.29 | 0.234 | 0.123 | 0.0579 | -0.033 | -34.7 | 0.242 | -0.014 | -0.00541 |
| EC ($\mu g$/std $m^3$) | 1.89 | -0.53 | -28 | 3.28 | 0.612 | 0.295 | -0.787 | -41.6 | 3.7 | 0.448 | 0.195 |
| OA ($\mu g$/std $m^3$) | 156 | -146 | -93.4 | 420 | 0.612 | 0.0174 | -147 | -94.2 | 423 | 0.472 | 0.0122 |
| Sulfate ($\mu g$/std $m^3$) | 0.791 | -0.116 | -14.7 | 0.676 | 0.415 | 0.184 | -0.214 | -27.1 | 0.728 | 0.322 | 0.13 |
| $NH_4^+$ ($\mu g$/std $m^3$) | 1 | -0.591 | -59.1 | 0.931 | 0.767 | 0.351 | -0.615 | -61.5 | 0.956 | 0.729 | 0.359 |
| Nitrate ($\mu g$/std $m^3$) | 1.7 | -0.56 | -32.9 | 1.47 | 0.805 | 0.613 | -0.634 | -37.2 | 1.59 | 0.774 | 0.599 |
| AOE (/Mm) | 391 | -350 | -89.3 | 994 | 0.688 | 0.027 | -357 | -91.1 | 1010. | 0.532 | 0.0152 |

Table 3, same as Table 2 except for the wildfire affected flight segments.



a) GFS mean 2m temperature bias (K), 08/2019     b) WRF mean 2m temperature bias (K), 08/2019

c) GFS mean 2m specific humidity bias (g/kg), 08/2019     d) WRF mean 2m specific humidity bias (g/kg), 08/2019

e) GFS mean 10m wind speed bias (m/s), 08/2019     f) WRF mean 10m wind speed bias (m/s), 08/2019

Figure 1. GFS and WRF surface meteorological biases for METAR (METeorological Aerodrome Report) stations averaged over August, 2019



a)

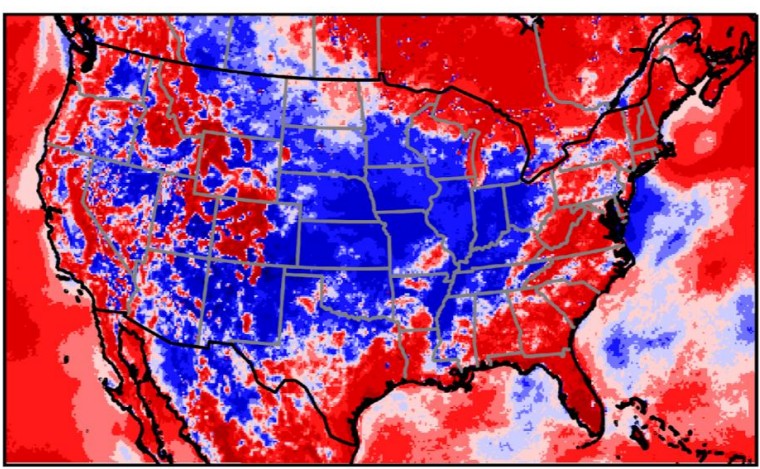

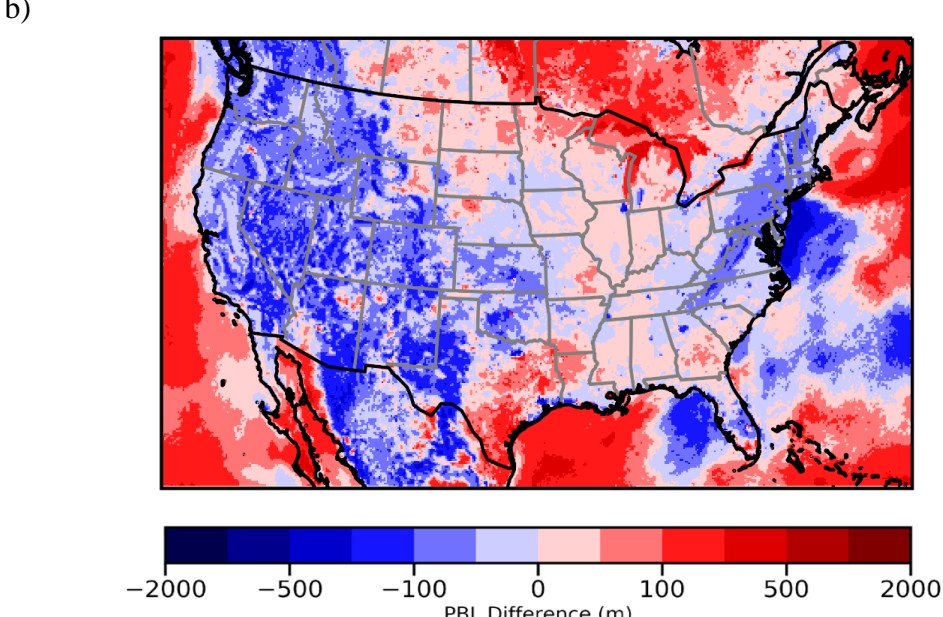

Figure 2. Monthly mean PBL height difference (GFS-WRF) for daytime (a) and nighttime (b), August, 2019.




Figure 3. The WRF and GFS time-series comparison over AIRNow stations over the U.S. West and Northeast for 2m temperature (a, b), 10m wind speed (c, d), and PBL height (e,f).





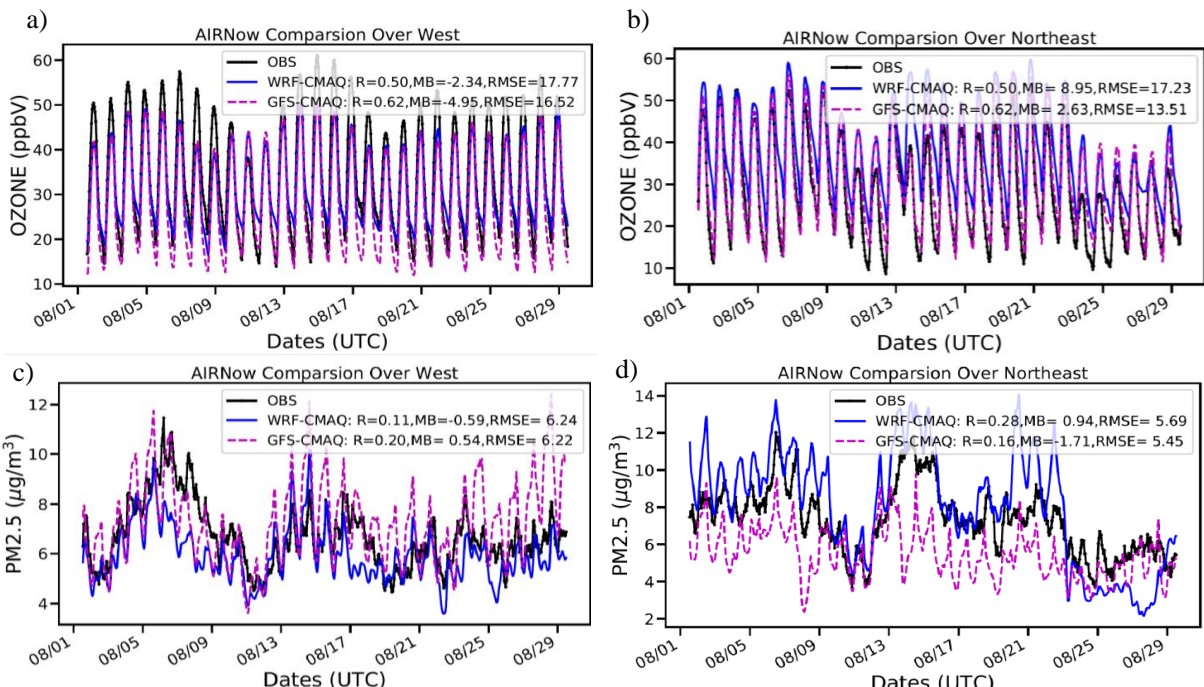

Figure 4. Same as Figure 3 but for ozone (a, b) and PM2.5 (c, d).





Figure 5. Monthly mean surface ozone predictions by GFS-CMAQ (left plots) and WRF-CMAQ (right plots) for daytime (top plots) and nighttime (bottom plots) compared to the corresponding AIRNow observations, August, 2019





Figure 6, same as figure 5 but for surface PM2.5





Figure 7. Modeled meteorological variables compared with observations for the DC-8 flight on 22 July, 2019 (b to f). The plot a shows the flight path colored in altitudes above sea level with UTC time in red text. Base map credits: © OpenStreetMap contributors 2022. Distributed under the Open Data Commons Open Database License (ODbL) v1.0.




a)  GFS-CMAQ predicted vertical velocity along the DC-8
flight path, 22, July, 2019

b)  WRF-CMAQ predicted vertical velocity along the DC-8
flight path, 22, July, 2019

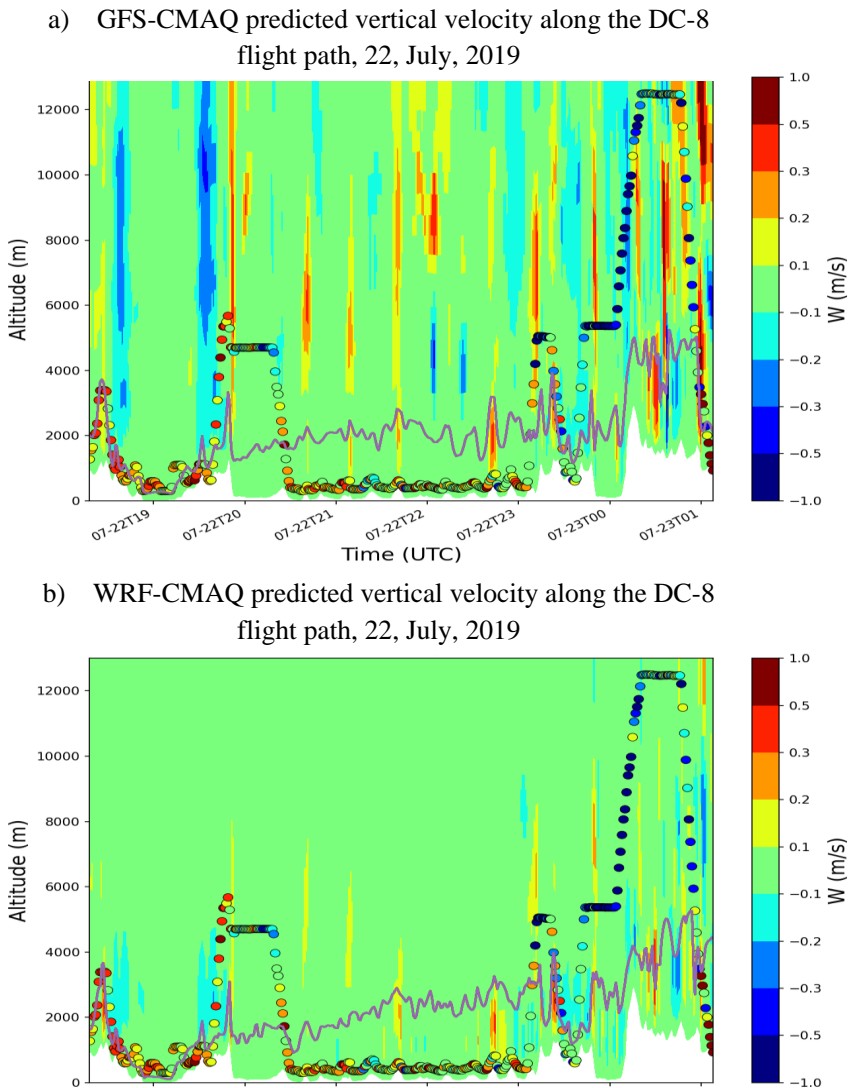

Figure 8, Curtain plots of the vertical velocity (W) predicted by GFS-CMAQ (a) and WRF-CMAQ (b) along the
DC-8 flight on 22 July, 2019. The colored dots showed the DC-8 measured vertical velocities. The solid lines
showed the predicted PBL heights of these two models.







Figure 9. Model predicted chemical concentrations compared with observations along with the DC-8 flight on 22 July, 2019





Figure 10. The DC-8 flight path (a), and model-observation comparisons for O$_3$ (b), CO (c), NO$_2$ (d), submicron organic aerosol (OA) (e) and aerosol optical extinction coefficient (AOE) at wavelength of 550nm (f) on 06 August, 2019. Base map credits: © OpenStreetMap contributors 2022. Distributed under the Open Data Commons Open Database License (ODbL) v1.0.



a) DIAL-HSRL aerosol backscatter coefficient (ABC) at 532nm wavelength along the DC-8 flight path, 6 Aug., 2019

b) DIAL-HSRL cloud screened ABC at 532nm wavelength along the DC-8 flight path, 6 Aug., 2019

c) GFS-CMAQ predicted AOE@550nm (/Mm) along the DC-8 flight path, 6 Aug., 2019

d) WRF-CMAQ predicted AOE@550nm (/Mm) along the DC-8 flight path, 6 Aug., 2019

e) GFS-CMAQ predicted RH (%) along the DC-8 flight path, 6 Aug., 2019

f) WRF-CMAQ predicted RH (%) along the DC-8 flight path, 6 Aug., 2019

Figure 11, The Differential Absorption High Spectral Resolution Lidar (DIAL-HSRL) retrieved aerosol backscatter coefficient (ABC) at 532nm wavelength in unit /km/steradian (a) and cloud screened one (b); curtain plots of the AOE (b, c) and relative humidity (RH) (d, e) predicted by GFS-CMAQ (left) and WRF-CMAQ (right) along the DC-8 flight on 06 August, 2019. The colored dots showed the corresponding measured values. The solid lines showed the predicted PBL heights of these two models.