# Peer review of "Evaluation of the NAOFC Driven by the NOAA Global Forecast"

_EGUsphere, 2022_

## Author Response (AR1)

**Dear Editor and Reviewers**

**Thank you for your comments. We followed your comments to revise this manuscript.**

**Besides the literature changes suggested by you, we replaced the Figure 11 and added a new figure (Figure S2) which is about the vertical velocities in original GFS/WRF and those diagnosed in CMAQ.**

**Please see the revised manuscript for detail.**

**Thank you again for your comments**

**Answer to Review #1**

**Thank you for your valuable comments. The manuscript is revised accordingly, including a new Figure S2. Here are the answers to your comments.**

*This article provides an incremental step forward compared to the Campbell et al 2022 paper. Campbell et al provides a huge step forward for the regional air quality modeling field. A major limitation for regional models has been coupling to existing available meteorology. The Campbell et al 2022 is a great addition that includes a similar NACC-CMAQ and GFS-CMAQ comparison, and this paper helps to strengthen evidence that the resulting modeling is credible. If I am correct, the previous Campbell et al. comparison did not isolate meteorology differences. This paper uses the NEIC 2016v1 emission inventory for both models (as well as GBBEPx and BEIS), which allows for a more clear isolation of meteorology. The meteorology still includes both physics-parameterization, scale, input and interpolation differences. The isolation of met is definitely a strength. The paper uses validation against FIREX-AQ and one month against surface observations. The weakness in this comparison is the focus on only summer months, which highlights ozone performance more than PM2.5.*
*Response:*
*Overall this is a good paper that characterizes model performance of an important configuration (GFS-CMAQ) and compares it to a more common application (WRF-CMAQ). Perhaps my one disappointment was that the time period for AirNow evaluation was very short and may not highlight issues under a variety of conditions where the model will be applied. I support the publication of this manuscript. Hopefully, minor notes below can be incorporated.*

➢ Thank you for your encouraging. This manuscript is trying to expand our previous paper by comparing this method with the prevailing WRF-CMAQ system. The overall result show these two systems are similar, and their difference is mainly related to the meteorological models' dynamic/physics, not the coupler. So, the interpolation-based coupler is useful when the driving meteorological variables are sufficiently available. Our previous paper (Campbell et al, 2022, https://doi.org/10.5194/gmd-15-3281-2022) included longer time verification.

*Pg 7, Ln 4: The 127 layers in GFS are not comparable to the 35 in CMAQ. As noted in your Table 1, GFS uses 127 to reach 80km where CMAQ reaches 100 hPa, which is more like 17km in the international standard atmosphere. It would be more interesting to note the number of layers that are likely to affect the near-surface winds.*

➢ It is true that the GFS has much more vertical layers than WRF. However, GFS has much higher heights, and they has similar vertical layers below 1km. The GFS meteorology was also collapsed into the 35 layers to drive CMAQ. So they are comparable in certain extent, as shown in Tables 2, 3 for altitude below 3km. Our other paper (Campbell et al. 2022) included the comparison with the previous version: GFSv15-CMAQ, showing that the GFSv16-CMAQ and GFSv15-CMAQ could have significant difference over certain region, mainly due to their difference physics. We added some words about that.

*Pg 7, Ln 37: for[ ]10*
➢ Changed

*Figure 2: Ideally, these would be based on local times. The 6UTC is about 1am on the east coast while the west coast is more like 10pm. And 18UTC is about 1pm on the east coast and 10am on the west. I know the authors are aware and likely have already considered the implications based on the PBL rise, but the reader won't be fully aware of the differences in model PBL development rates. The sharp rise and collapse noted by the authors raises questions. For example, the geographic differences could have something to do with the rates of rise and drop rather than the ultimate depths.*

➢ Yes, it is true that the selected times may not represent around-noon and midnight situations across the CONUS domain, since it is hard to find one-fit-all time for the 4 time zones. Figure 2 only shows the normal daytime (nighttime) monthly-mean situation after sunrise (sunset), and 18UTC/06UTC are not in the transition time ranges for the sharp rise and collapse of PBL around sunrise/sunset. So these selected times avoided the PBL's fast-change time. It is true that the PBL spatial variations are related to regional geographic differences. We added some explanation on page 7 about the issues.

*Figure 3 is very interesting!*

➢ Thank you for this comment

*Pg 8, ln 24: same as pg 7, ln 4*

➢ Changed

*Pg 10, ln 24-26: How does the resolution you are showing here compare to the resolution used in the CMAQ modeling? Was the NACC output done at hourly resolution?*

➢ As mentioned in the manuscript, we use one-minute averaged flight data, and the models' hourly 12km outputs are spatiotemporally interpolated to the flight paths for comparison. Yes, NACC output was in hourly resolution.

*Pg 11, ln 2-4: Can you expand a bit here? Is the CMAQ-diagnosed W just as appropriate for the NACC interpolated values?*

➢ Expanded by adding Figure S2 and corresponding discussions.

*Pg 11, ln 15-17: Is the ozone here VOC-limited? Or, why do you think this when the NOx is underestimated from 20:30Z to 23Z and ethene (a highly reactive VOC) is overestimated? Underestimated CO can actually increase the yield of HO2 per OH reaction.*

➢ Over some segments of the flight 07/22 with high NOx, ozone tends to VOC limited. The NOx underestimation for 20:30-23UTC could be seen in Figure 9, while ethene was slightly overestimated by about 0.2 ppbv, and ethane was underestimated by about 1 ppbv. We removed CO in that sentence.

Pg 12, ln 12-14: Could the high altitude underpredictions be due to instrument interference from CH3O2NO2? (e.g., Browne doi:10.5194/acp-11-4209-2011)

➢ Good information. I am not sure whether CH3O2NO2 caused this issue. However, the NO2 instrument has detection limit around 0.01 ppbv according to. https://airbornescience.nasa.gov/sites/default/files/documents/NOAA%20NOyO3_SEAC4RS.pdf . We added this information.

*Figure 11: The black outlines of the dots makes it impossible to see the values when at a constant altitude. Can you remove the outline or widen the plots to provide more visibility?*
➢ These plots were replaced and the observations became more visible.

*Pg 13, ln 29: nitpicky, but 25UTC should be 01UTC Aug 7*
➢ Added the 01UTC

*Pg 14, ln 31: I am interpreting non-fire events as all non-fire times. So, these aren't really events. In that way, I would avoid calling these "2019 events" since you are talking about overall statistics. Similarly, the concentrations of SO2 you are seeing are quite low -- the mean biases are fractions of a microgram or ppb. Based on these magnitudes indicating ambient values, it seems plausible that the this could indicate chemical lifetimes or deposition errors too? BTW, most power plants have continuous emission monitoring, which leads to more certain emissions.*
➢ Changed the "non-fire events" to "flight segments without fire influences". For SO2 underestimation, you are right that power plant emissions could be the issue. To emulate the forecast behavior, we did not use continuous emission monitoring data which is available after the events, but just the original NEIC 2016 point source inventory. Some sources supposed to shut down in the original inventory might still emit pollutants during the flight observations, leading to the disagreement.

*pg 15, ln 1-23: The observed NOz is about 1.63 ppb while the sum of measured HNO3 and PAN is 0.69 ppb. True the PAN bias is very low (-0.22 or -0.25 ppb), but clearly this is disproportionate to the NOz low-bias (-0.96 ppb). I would have liked to see a more clear discussion of the role of nitrate in NOy and therefore in NOz. At least some fraction of particulate nitrate is usually included in the NOy measurement (doi: 10.1021/es501896w). The model NO3 is low biased by about 1 microgram/m3, which when converted to std-ppbv is a non trivial fraction of the NOz bias. It would be nice to have a more clear discussion of what contributes to the NOz underestimation.*
➢ The discussion was expanded. NOz species includes inorganic, such as HNO3, HONO et al and organic: PAN, MPAN, OPAN, RNO3 et al. The organic NOz, such as PAN, was underpredicted, associated with underestimation of certain VOC species. The NOy observation could include some particulate nitrate, but should be very limited. According to Ryerson et al (1998, https://agupubs.onlinelibrary.wiley.com/doi/10.1029/1998JD100087), in this NOy instrument, "aerosol transmission is not characterized, but inlet design and orientation probably discriminates against the majority of aerosol by mass". The particulate nitrate ion was also underestimated, but its precursor HNO3 was overestimated. As discussed in the next paragraph, this issue should be related to the underestimation of cations, like $NH_4^+$, which caused the shift of gas-aerosol equilibrium partition shift of the nitrate ions

*Pg 15, ln 25-30: Can you discuss where the errors may originate like you did with NOz and NH4, and nitrate?*
➢ We expanded some. Please check the revised manuscript.

**Thank you again for your comments.**

**Answer to Review #2**

**Thank you for your valuable comments. The manuscript is revised accordingly, including a new Figure S2. Here are answers to your comments.**

*This manuscript presents a comparison of predictions from the CMAQ model driven by meteorological fields derived from the FV3-GFSv16 (or GFS) and WRF models for periods coinciding the 2019 FIREX-AQ field campaign. The GFS-CMAQ system comprising of CMAQv5.3.1 and the FV3-GFSv16 now constitute the operational National Air Quality Forecast Capability. Consequently, a study assessing the performance of the system relative to both predictions from another modeling system (in this case the widely used WRF-CMAQ configuration) as well as multi-species measurements from a field campaign would be of interest for establishing the efficacy of the recently updated NAQFC system. While the manuscript presents a significant amount of work in terms of simulations with the two modeling systems and comparisons with airborne measurements taken during the FIREX-AQ campaign, and for most part shows similar performance for the GFS-CMAQ and WRF-CMAQ models, the execution of the analysis in the current form in my assessment did not present in a compelling manner either the impacts of the different meteorological drivers on the predicted air quality or the ability of the NAQFC improvements in capturing the effects of fire emissions on air quality. Some of this perhaps results from the study design trying to cover too much ground. For instance, uncertainties in estimating fire emissions (magnitude, composition, placement (vertically and burn area)) confound isolating meteorological differences from those resulting from emission uncertainties. Comparisons with high time (and space) resolution measurements are confounded by limitation in model grid to reasonably isolate meteorological difference impacts. Similarly, use of data assimilation (strong nudging) also likely reduces the differences between the GFS and WRF fields making it difficult to identify impacts associated with either physics (PBL, cloud representation), or computational differences (interpolation vs. native grid). Perhaps a clearer articulation of the study objectives and closer linkage of the analysis with the objectives would help improve the overall usefulness of the manuscript. Also, many of the inferences conveyed by the analysis could benefit from additional substantiation. The following suggestions are offered that may help address these shortcomings:*

➢ Thank you for your comments. It is true that this study inevitably includes some general discussions for various issues of air quality modeling, though it aims on inter-comparison of interpolation-based FV3-CMAQ and traditional WRF-CMAQ. First, as mentioned in the conclusion, these two approaches yielded overall similar results, especially in the comparison with the aircraft data. Second, their difference appeared in some events, regions, mainly due to their different physics etc. In order to minimize this influence, the nudging method used in WRF is to reduce this physical difference, though it is not strong enough. Third, they could have common biases, due to emissions etc. We did include some analysis and discussion for these common issues of air quality modeling, which may cause some feeling of complexity. In fact, these two approach showed the similar biases under most circumstances, implying the difference between interpolation method vs WRF is the reason of these biases. We made some changes to articulate the main point.

1. *I find the use of the terminology WRF-CMAQ downscaling (in the title and manuscript text) to be somewhat confusing in context of what is presented. In my view "downscaling" is methodology typically used to translate coarse grid information to finer resolutions with a dynamical model but without any other observational assimilation. Why not just refer to it as WRF-CMAQ configuration?*

➢ We changed it in the title and some places in the manuscript. Here the WRF's role is exact as you mentioned: transform the global GFS data to finer grid to drive CMAQ without observational assimilation. The WRF's nudging was toward GFS, not toward observations.

2. *It would be useful to explain in a little more detail the differences in data assimilation approaches used in the FV3-GFS and WRF simulations. Since this is a retrospective analysis, it does appear from Table 1 that WRF did utilize data nudging. How often were WRF runs initialized? Does the 4D-IAU utilized in GFS also include the observations that were incorporated in the WRF nudging? How often were the GFS runs initialized? Some discussion of the impacts assimilation (nudging) approaches in the two models on meteorological fields and any differences would be useful.*

➢ We expanded some. The GDAS assimilation was used for GFS re-initialization daily. WRF ran continuously without re-initialization, but with nudging toward the GFS every 6 hours for horizontal wind, temperature and humidity. Since WRF was nudged toward GFS, WRF did not use any observation directly, but depended on GFS for its lateral boundary condition and nudging target. From the comparisons conducted afterward, the nudging effect seemed work quite well in elevated layers, and WRF and GFS yielded similar results compared to aircraft data. However, these two models showed some difference near surface due to their different physics.

3. *Page 4, L11-12: Since the NACC uses bilinear interpolation, why is the interpolation of the scalar variables more straightforward and relative to what?*

➢ It refers to the interpolation from GFS A-grid to CMAQ A-grid, for most scalar variables, such as temperature, humidity etc. We added the statement.

4. *Page 5, L4-10: Why were the WRF model top and vertical grid structures chosen to be so different from those of the GFS? Does that not also introduce another source of differences in the predicted meteorological fields and confound interpretation of differences arising due to different model structures vs. interpolation to a coarse vertical layer structure? What is the vertical structure of the CMAQ model deployed in the NAQFC?*

➢ It is mainly because the NAQFC's CMAQ uses the 35 vertical layers, and WRF follows that vertical setting. NAQFC is designed to forecast near-surface air quality, and has relatively dense vertical layers (similar to the 127-layer GFS) below 1km. You are right that the different vertical resolution could introduce some difference, and the nudging method was used to reduce it. It should be noted that WRF's vertical coordinate also differs from GFSv16's sigma-P hybrid coordinate, and they can not be exactly same anyway. This purpose of study is not exploring various WRF setting to yield the nearest results to GFS. Its main focus is about whether using interpolated meteorology to drive CMAQ is valid, compared to a typical WRF-CMAQ setting: a 127-layer WRF for CMAQ application would be very rare. If this approach is valid and does not yield abnormal results, we need not run WRF at all, which can save computing time for CMAQ to forecast extra 24 hours during the

operation. The comprehensive comparison with aircraft/surface measurement showed that the two models have overall similar meteorological and air quality predictions. They have some difference depending on regions and events, but neither of them is overall superior to the other. We revised some conclusions to highlight this point.

5. *P5, L23: what does a "novel" inline dust model imply? As written, it is not readily apparent to the reader what novelty the FENGSHA approach provides for dust emissions relative to other approaches. Please expand briefly to better convey the novel aspects of the system, especially those that pertain to forecasting.*
➢ Changed and expanded.

6. P5, L31: What does "total organic aerosol" imply especially in context of emissions? Did the authors imply primary organic aerosols?
➢ Yes, you are right. Changed

7. *P5, L36-40: The description of scaling of wildfire emissions could be improved. Why are only fires west of 110W assumed to be long lasting? How is the model grid cell fraction (am assuming of burned area) determined?*
➢ Added the explanation. "As historic statistics shows that most fires (>95%) in east of 110°W last less than 24 hours". For burning area, "Burning area could be highly uncertain as GBBEPx data do not have this information. One grid cell could have multiply fires and some big fire could appear in several grids. Here we carry the previous NAQFC's method, and apply a constant ratio: 10% of the grid cell, as the burning area (Pan et al., 2020) according to Rolph et al (2009)."

8. *Section 3.1: Did the authors compute performance statistics for meteorological fields from both the native GFSV16 as well as the fields interpolated to the CMAQ grid? Were these comparable? It may be useful to state so.*
➢ It uses the interpolated GFS meteorology for comparison, and it is very consistent with the original GFS data. We expanded the statement

9. *P7, L8: It was not apparent to me what "produce synoptic scale body forces" implies. Please clarify.*
➢ Changed to "influence the synoptic-scale dynamics". Please refer to Kim et al. (2003) for detail: "Atmospheric gravity waves are one such unresolved process. These waves are generated by lower atmospheric sources, e.g., flow over irregularities at the Earth's surface such as mountains and valleys, uneven distribution of diabatic heat sources associated with convective systems, and highly dynamic atmospheric processes such as jet streams and fronts. The dissipation of these waves produces synoptic-scale body forces on the atmospheric flow, known as "gravity-wave drag" (GWD), which affects both short-term evolution of weather systems and long-term climate."

10. *P7, L13-20: I found it difficult to quantitatively infer the PBL differences from the color scales depicted in Figure 2. It may be useful to also include histograms of the differences. Are the monthly means based on all hourly data or only daytime values? Are monthly mean differences of the order of 500-1000m not too large? Would these not result in more pronounced differences in predicted ambient concentrations?*

➢ Yes, the color scheme may be difficult for quantitatively seeing daytime PBL difference. We tried several color schemes, but all of them have some drawbacks. Most qualitative color schemes have issues to highlight the positive and negative difference, which is very important. Figure 3 showed quantitative comparison of PBL over two regions. All the curtain plots (Figure 8 and 11) also included PBL information. The monthly mean use two daily times to represent daytime (18UTC), and nighttime (6UTC). Yes, 500-1000m PBL difference is not small, and this meteorological difference will affect the corresponding air quality predictions. Unfortunately no reliable PBL observations are available to verify these PBL and show whether WRF or GFS has bigger PBL prediction bias. This result shows that different model with different physics could result in up to 50% or higher PBL difference.

11. *Figure 3: Significant differences in WRF wind fields in the West and Northeast are shown for the period after 8/21. Why do these arise if winds were nudged to observations? The discussion on page 8 suggests that "some WRF setting" leads to large bias for storm weather, but it is not clear what these settings are and what causes the error? It would be useful to expand the discussion on page 8.*
➢ The nudging was toward GFS every 6 hours, not to observations. However, it is still insufficient to offset the 10-m wind bias. We expanded it. However, since this manuscript's focus is about the air quality modeling issues, we did not dive into the detail about WRF's configuration for storm weather prediction.

12. *P8: should "northwest" on L28 be "west"? L30: perhaps "shallower PBL" is better description than "thinner PBL"*
➢ changed

13. *P8, L32-33: At night, is the stable surface layer not decoupled from the residual layer, so that there is not much exchange anyway?*
➢ Thank you for this suggestion. Changed

14. *P9: demonstrating the impacts of systematic differences in PBL heights and wind fields between GFS and WRF on the eventual PM2.5 distributions is very useful. However, I am not sure I fully understand the discussion. The GFS nighttime PBL heights appear to be lower (Fig 2b), but so do the PM2.5 (Fig 2c). Would both shallower PBL and weaker winds not result in higher PM2.5 within the stable boundary layer in GFS-CMAQ? While it is possible that lack of transport, results in lower downwind PM2.5 concentrations, would the lower PBL depicted in Fig2 not result in higher PM2.5 at least in the source regions at night?*
➢ Yes, the lower PBL corresponds to higher PM2.5 over source region. GFS-CMAQ's nighttime PBL is lower (Fig 3e), and so its PM2.5 should be higher (Fig 4c) if both models have same background, like over the U.S. West. The GFS-CMAQ's lower PM2.5 over the Northeast was mainly due to the different background/upstream concentrations.

15. *Would the impact of these large differences on predicted air quality perhaps be better captured through comparisons with more widespread surface observations (of ozone and PM2.5 and constituents) than just limited flight segments, especially given the uncertainties in the estimated fire emissions? Hourly surface measurements (say from the AQS network) may also help better delineate the impacts of any systematic diurnal*

*differences in WRF and GFS predictions on the eventual atmospheric composition predictions.*

➢ The section 3 included the comparison with surface observations. It is true that the flight observation may not have widespread area coverage. However, the aircraft observation shows the comparison in the elevated layers with comprehensive chemical species. It provided a different view angle. Both surface and aircraft comparisons are useful. Our previous paper (Campbell et al, 2022, https://doi.org/10.5194/gmd-15-3281-2022) mainly uses surface observations for comparisons.

16. *Figure 7: How well did the native WRF and GFS vertical wind velocities compare with the measured ones? Also, it appears that the re-diagnosed W in GFS-CMAQ show greater variability than those in WRF-CMAQ, especially at higher altitudes – would the authors know what may cause that? If it is due to interpolation, does it suggest an issue with the interpolation scheme employed for the GFS fields? Some additional discussion would be useful in context of the variability in the observed W fields.*

➢ We added a new figure (Figure S2) and corresponding discussion for the W issue. The stronger W of GFS-CMAQ compared to WRF-CMAQ was brought by the original models: GFS vs WRF, instead of the interpolation issue.

17. *P11, L16-17: The sentence should be reworded a bit – the relationship between underestimation of CO and NOz is not apparent to me.*

➢ Revised

18. *P12, Figure 10 discussion. The model CO values do not appear to deviate too much from their "baseline". Is this because of underestimation of fire emissions or possibly due to the model fire plume being displaced relative to the flight path?*

➢ The CO underprediction happened in both elevated layers (around 19UTC) and lower altitudes (21UTC). So, both emission and plume rise could have issues. Another possible issue is the dilution due to the model resolution, as we discussed later.

19. *P12, L17-19: does "background" represent regional average values or the amount of ozone from outside the domain?*

➢ It should be the regional background, not lateral boundary condition. Revised

20. *P12, L21-23: I am curious why modeled NO2 as opposed to CO is used as an indicator of fire location – would CO not be a better tracer? How were the column averaged NO2 and ozone observations (filled circles) shown in Fig S3 computed?*

➢ Both $NO_2$ and CO can be used as fire indicator. We use NO2 here mainly as it has shorter lifetime, and is less affected by regional background. Another purpose of using NO2 is for the next discussion about ozone.

21. *P12, L31: flight path missing the locations of the modeled peak values is a somewhat awkward way to suggest that the model and observed plumes were misplaced. Consider rewording this sentence.*

➢ Reworded

22. *The axis labels in the sub-figures of Fig 11 are extremely small and illegible.*

➢ Changed

23. *Tables 2 and 3: Please explain how the data was segregated into fire and no-fire bins used for these statistics.*
➢ Added. FIREX-AQ aircraft data has smoke flag, which is used to distinguish fire and no-fire bins.

24. *P14, L19-21: Please explain why accounting for warm bias during non-fire impacts would be influenced by the lack of using wildfire heat content in the energy balance equations of the meteorological models. Would the warm bias not get exacerbated?*
➢ The fire heat flux was not considered in either of the meteorological models, which would lead to the meteorological prediction biases. This simple comparison can not tell the exact quantitative bias due to this ignorance.

25. *P14, L38-39: I was surprised that the SOx bias was attributed to the SO2 point source inventory? Do the emission estimates used in these simulations not use the continuous emissions monitoring data for point sources which provide accurate constraints for SO2 emissions from point sources?*
➢ This study emulates the forecast situation, where the continuous emissions monitoring data of power plants are not available in real time. So we just used the original NEIC 2016 point source inventory. Some sources supposed to shut down in the original inventory might still emit pollutants during the flight observations, leading to the disagreement.

26. *Tables 2 and 3: I was curious why the sum of the observed NOx+NOz is not equal to the observed Noy values shown in these tables? It may be due to differences in the averaging but would be useful to verify and explain.*
➢ It is due to sampling number issue of these three species. NOx and NOy observations have different missing data, and NOz is calculated when both observation are available at a certain sampling time. Added the explanation.

27. *P15, L1-4: While PAN is underestimated, HNO3 is not – do the PAN underestimates fully explain the NOz underestimation? What other species constitute NOz in the analysis presented?*
➢ This part is revised. NOz species includes inorganic, such as HNO3, HONO et al and organic: PAN, MPAN, OPAN, RNO3 et al. We use PAN as an example to represent organic NOz NOz, which was underpredicted, associated with underestimation of certain VOC species.

28. *P15, L11: would be useful to provide a reference for the "our other analysis".*
➢ Revised.

29. *P15, L14: it is not clear what VOC "speciation issue" in NEIC2016v1 is implied here - please elaborate.*
➢ Changed

30. *P15, L18: what size range does the "submicron ammonium" refer to and how were the corresponding model values estimated?*
➢ Added that information. Submicron aerosol refers to the fine aerosol with diameter $< 1\mu m$. CMAQ model outputs aerosol fraction ratios in different size bins, such as diameter $< 1\mu m$

(submicron), diameter < 2.5 μm (PM2.5). Those information are used for calculate submicron aerosols

31. *P15, L22: I was curious what chemical pathway in the model converts HNO3 to organic nitrates? Does the gas-phase chemical mechanisms employed include such conversions?*
➢ Thanks for this finding. It should be NOx and inorganic NOz (such as NO3), not HNO3.

32. *P16, L2-5: One way to isolate the effects of resolution vs. emission errors could be to examine ratios of model and observed species associated with fire emissions, since the ratios will not be impacted by systematic artificial dilution effects of grid resolution.*
➢ Yes, that was our first idea in this analysis. We tried to use relatively reliable fire tracer, such as CO, and base species, and analysis other species' ratio to CO. We encountered two issues. One is the CO background, since most fire areas are adjacent and it was hard to isolate a specific fire spot's CO emission, especially for small or medium fire. Another issue is the different lifetime of pollutants, which highly depending on the sampling distance from the fire sport. Some VOCs, such as aldehydes, can be formed secondarily, which should be treated with caution.

33. P16, L6-7: Given that there is not much difference between observed O3 values of 58 and 56ppb, I am not sure one can discern systematic titration effects conveyed in this discussion. Similarly, the O3 enhancements suggested in downstream areas (L11) are not apparent in the comparisons presented. Some additional discussion and substantiation of what is being conveyed would be useful.
➢ The $O_3$ titration effect refers to Figure S4 and related discussion, instead of Tables 2, 3. The mean fire events include all flight segment with fire influences, near sources and further downstream. The titration effect can only be seen near fire sources. We revised.

34. *P16, paragraph starting L30: The discussion contains many sentences that are somewhat vague and should be rewritten. As examples: (i) why only refer to advection as horizontal? Similarly, does the vertical transport imply only cloud mixing? (ii) What is the issue mentioned by Qian (2020)? The physical significance of the issue should be mentioned. (iii) How does one quantitatively ascertain that the differences between the GFS and WRF physics have larger impact on the meteorological fields than the "meteorology driver" methodology, since no interpolation was used in recasting the WRF fields for CMAQ? Some elaboration would be useful to support the statement.*
➢ This paragraph is revised. (i) changed to "horizontal/vertical wind fields". (ii) It refers to the common model problem of neglecting the irrigation impact, mentioned in section 3.1 (iii) since this study focus on air quality modeling, we did not dive into the detail how the GFS and WRF physics differ. However, from the limited analysis in this study, it is evident that the impact of different physics is more significant than that of interpolation vs native grids.

35. *P17, L37-40: Given that the FV3-GFS and CMAQ grid structures are different, what option other than interpolation is available? I am not sure I see what specific advantage of the NACC is being conveyed here. It is also not clear what is implied by "it avoids running another model as WRF" since the choice of the meteorological driver appears to be user driven. It seems the NACC was created for a specific purpose of translating the*

*GFS meteorological fields and interfacing with CMAQ and as such does not preclude use of other models such as WRF but just provides an option. It is also not clear what is meant by the statement that the NACC is "faster and more consistent with the original meteorological driver"? Relative to what? What does consistency imply in this context? Also, as indicated earlier, I feel that the "downscaler" terminology for WRF unnecessarily introduces additional confusion.*

➢ For the general regridding method, https://earthsystemmodeling.org/regrid/#regridding-methods has 6 options. Currently NACC uses bilinear (for temperature etc) and nearest neighbor (for landuse etc). As mentioned here, NACC has two advantages over WRF for operational air quality forecast. One is the speed: in current NCEP operational NAQFC system, NACC only takes less than 5 minutes to process 72-hour data, which saves enough time for CMAQ to forecast extra 24 hours. Another advantage is its consistency with original GFS data, since WRF's own physics would alter the meteorology anyway even with nudging. It can also save the significant effort of tuning WRF.

➢ "consistent with" original GFS data

➢ removed "downscaler"

*36. P18,L12-13: "enough variables to drive CMAQ with other supplied data" is awkward and should be reworded.*

➢ reworded

*37. P18, L15-25: I found this discussion to be somewhat wishy-washy with no clear conclusion or indication of next steps.*

➢ This part is revised. As mentioned above, the interpolation approach provided an alternative method to drive CMAQ other than WRF-CMAQ. Both methods have its pros and cons. We are working on cloud-based NACC processor for community usage, which is the next step.

**Thank you again for your comments.**

**Answer to Editor's Comments #3**

*"Technical comments that may affect the quantitative results: I want to point out that airnow surface observations they used to evaluate the models have a much shorter latency than the more widely used AQS data (cited in review #2 report). While airnow data may be more suitable for being assimilated to improve initial conditions of model forecasts (which is not relevant to this paper), they are not the best evaluation data for retrospective analysis like this. Airnow raw data are without any quality control, seemingly unreliable values are found sometimes, and unlike AQS, no measurement detection limits (depending on measurement approach) are provided. If airnow data will continue to be used, at least a rigorous approach to screen the data must be developed and validated, referring to additional information. Also, more details on how the observations are matched with the model results need to be provided. Specifically: often, multiple surface observations with huge variability fall within a given model grid cell, and were they averaged before being compared with the model results? In other words, how were horizontal representativeness error being handled? Over the complex terrain regions like the western US where some of their analysis is focused on, the model terrain is often unrealistic, while observations at mountain sites may reflect both boundary layer and free tropospheric air influences via downslope and upslope flows, did they always extract the model outputs from the "surface" level at the observation locations?*

➢ Thank you for asking this important verification issue. You are right that AQS, the quality-controlled AIRNow, is more reliable, and should be used whenever it is available. In fact, we had verification tool to process both AQS and AIRNow https://monet-arl.readthedocs.io/en/stable, and compared the AIRNow with AQS verification for other periods. For ozone and PM2.5, AIRNow and AQS agree very well in most scenarios, especially for data in recent 3 years. Occasionally AIRNow instrument over certain site during certain period may have issues, and we only found that this happened for PM2.5. This issue may arise if the comparison focus on certain problematic site. For most regional verification that include many sites, AIRNow and AQS data did not have systematic difference. We had screen filters in place to skip the problematic AIRNow data. This study uses AIRNow for surface verification mainly because our previous paper (Campbell et al, 2022, https://doi.org/10.5194/gmd-15-3281-2022) and NCEP operational verification use it. Our comparison is pairing data using model-to-obs approach: interpolate model data to observation location. Since the surface observation's heights are within the height of the model's lowest layer, Yes, the data from model's lowest layer were extracted for surface comparisons. For aircraft data comparison, the model data were 4-D spatiotemporally interpolated to the flight paths. About the surface sites' horizontal representativeness, since most AIRNow/AQS sites are near urban/suburban areas, they tend to represent more for urban/suburban. CMAQ does not have special treatment for the mountain-region meteorology, but depends on meteorological models for those information. WRF, as a common community model, and GFS as an operational model, have considered the topographic effect, though they may still miss some fine local features over complex terrains. As these possible-problematic mountain sites are not the majority in AIRNow stations, this issue unlikely changes the overall results of our regional verifications for ozone and PM2.5.

*Other comments: There are so many ways to set up and run WRF, and thus WRF can be viewed as more than one single model. Except that GFS and WRF-ARW dynamic cores are*

*different, WRF can be set up similar to and dependent upon GFS. Conclusions based on some comparisons of CMAQ driven by GFS and one set of WRF for a selected period are bit hard to digest (if they are seen not somewhat misleading), as it is unclear what exact strengths and weaknesses of the meteorological models are being assessed relevant to air quality modeling. The surface and above-surface analyses from this paper are kind of detached from each other, and it is unclear which one would be more relevant to/matter more to daily forecasts.*

➢ Yes, we totally agree that WRF can not be viewed as one single model, as it can have hundreds of configuration combination. This issue is actually the main point of this study. This purpose of study is not exploring various WRF setting to yield the nearest results to GFS, nor finding strengths and weaknesses of the WRF configurations. Its main focus is about whether using interpolated meteorology to drive CMAQ is valid, compared to a typical WRF-CMAQ setting, which may not be perfect but can represent the normal configuration. If this NACC approach is valid and does not yield abnormal results, we need not configure and run WRF at all, which can save many efforts as well as computing time for CMAQ to forecast extra hours. Although this study includes many details of WRF-CMAQ and their influences, this main purpose is actually exploring another way to avoid so many complexities, but still yield reasonable air quality predictions. Surface and elevated comparisons look detached mainly because these two type observations are drastically different: long-term hourly surface data covers the whole CONUS with mainly two species (ozone and PM2.5), and minute-resolution 3-D aircraft data include comprehensive chemical species but for shorter time periods. These two type observations provided very different view angles for the model comparison. If GFS-CMAQ and WRF-CMAQ behave similarly, viewed from both angles, they are likely actually similar. The NAQFC focuses more on surface air quality, which is relevant to human exposure.

---

## Author Response (AR2)

**Answer to Review #3**

**Thank you for your valuable comments. The manuscript is revised accordingly, please see revised manuscript for detail. Here are the answers for your comments.**

*This paper compares the meteorological and chemical performance of two different ways to run CMAQ: the conventional WRF-CMAQ version and the latest GFSv16-CMAQ. Focus is given on August 2019 when the FIREX-AQ provided 3D observations for evaluation.*
*This paper is a revision and has been revised following comments from three reviewers. The authors have addressed the raised comments and I find the paper suited for publications. Below some minor comments that could help strengthen the paper.*
*1) Some improvements in the English language would be beneficial (e.g. the use of articles).*

➤ Thank you for your comments. We made some literature revisions. Please see revised manuscript for detail.

*2) I agree with the previous reviewers that one month is a fairly short time period and this limitation should be better acknowledged in the Introduction.*

➤ We add the acknowledgement for this study period in the introduction. "This study focuses on the period of summer 2019, and Campbell et al. (2022) evaluated the GFS-CMAQ for longer periods."

*3) Abstract: I suggest also stating the a comparison to operational surface met and chemistry observations is done*

➤ Added the statement.

*4) Section 2.1: Please also state what the temporal frequency of the driving met fields is*

➤ Added "hourly".

*5) Section 2.2: What is meant by the GFS levels were "collapsed"? Would interpolated be a better term?*

➤ Changed.

*6) Section 2.3: State how VOC emissions were derived for GBBEPx*

➤ Added

*7) Section 3.2: I am really puzzled by the sudden degradation of WRF 10m wind speed in the Western U.S. at the end of August. Can the authors elaborate on this?*
*Also related to this. Did the authors look whether WRF performance drifts over time? I would assume that despite nudging, an over one month long simulation might show a drift.*

➤ The WRF degrading issues are related to storm weather, tropical storm Erin near the east coast and tropical storm Ivo near west U.S., later August, 2019 (see Page 9, line 7-14). WRF's setting and configuration in this study are mainly for air quality study, not tuned for predicting storm weather.

*8) Page 9, Line 8: Did the authors confirm the higher NOx? If so, add at least ("Not shown")*

➤ Added the "not shown". The GFS-CMAQ's higher surface NOx at night near source regions can be confirmed (see plots below). The NOx difference is almost opposite to the PBL difference (Figure 2).

Monthly Mean NOx Difference (GFS/CMAQ - WRF/CMAQ) at 18UTC    Monthly Mean NOx Difference (GFS/CMAQ - WRF/CMAQ) at 06UTC

[Figure]

[Figure]

*9: Summary, Page 19: I see that the speed and less computing resources is a very valid argument. However, as for nudging I wonder whether the authors can also comment on what would be if WRF is run with observation nudging or meteorological data assimilation. Or at least acknowledge that this is also a possibility.*

➢ Added the acknowledgement "Nudging toward observations or including data assimilation may yield different results for the WRF run" on page 18.

Again. Thank you for your comments.